# Online Prediction at the Limit of Zero Temperature

**Mark Herbster**     **Stephen Pasteris**
Department of Computer Science
University College London
London WC1E 6BT, England, UK
{m.herbster,s.pasteris}@cs.ucl.ac.uk

**Shaona Ghosh**
ECS
University of Southampton
Southampton, UK SO17 1BJ
ghosh.shaona@gmail.com

## Abstract

We design an online algorithm to classify the vertices of a graph. Underpinning the algorithm is the probability distribution of an Ising model isomorphic to the graph. Each classification is based on predicting the label with maximum marginal probability in the limit of zero-temperature with respect to the labels and vertices seen so far. Computing these classifications is unfortunately based on a $\#P$-complete problem. This motivates us to develop an algorithm for which we give a sequential guarantee in the online mistake bound framework. Our algorithm is optimal when the graph is a tree matching the prior results in [1]. For a general graph, the algorithm exploits the additional connectivity over a tree to provide a per-cluster bound. The algorithm is efficient, as the cumulative time to sequentially predict all of the vertices of the graph is quadratic in the size of the graph.

## 1 Introduction

Semi-supervised learning is now a standard methodology in machine learning. A common approach in semi-supervised learning is to build a graph [2] from a given set of labeled and unlabeled data with each datum represented as a vertex. The hope is that the constructed graph will capture either the cluster [3] or manifold [4] structure of the data. Typically, an edge in this graph indicates the expectation that the joined data points are more likely to have the same label. One method to exploit this representation is to use the semi-norm induced by the Laplacian of the graph [5, 4, 6, 7]. A shared idea of the Laplacian semi-norm based approaches is that the smoothness of a boolean labeling of the graph is measured via the *"cut"*, which is just the number of edges that connect disagreeing labels. In practice the semi-norm is then used as a regularizer in which the optimization problem is relaxed from boolean to real values. Our approach also uses the "cut", but unrelaxed, to define an Ising distribution over the vertices of the graph.

Predicting with the vertex marginals of an Ising distribution in the limit of zero temperature was shown to be optimal in the mistake bound model [1, Section 4.1] when the graph is a tree. The exact computation of marginal probabilities in the Ising model is intractable on non-trees [8]. However, in the limit of zero temperature, a rich combinatorial structure called the Picard-Queyranne graph [9] emerges. We exploit this structure to give an algorithm which 1) is optimal on trees, 2) has a quadratic cumulative computational complexity, and 3) has a mistake bound on generic graphs that is stronger than previous bounds in many natural cases.

The paper is organized as follows. In the remainder of this section, we introduce the Ising model and lightly review previous work in the online mistake bound model for predicting the labeling of a graph. In Section 2 we review our key technical tool the Picard-Queyranne graph [9] and explain the required notation. In the body of Section 3 we provide a mistake bound analysis of our algorithm as well as the intractable `0-Ising` algorithm and then conclude with a detailed comparison to the state of the art. In the appendices we provide proofs as well as preliminary experimental results.

**Ising model in the limit zero temperature.** In our setting, the parameters of the Ising model are an $n$-vertex graph $\mathcal{G} = (V(\mathcal{G}), E(\mathcal{G}))$ and a temperature parameter $\tau > 0$, where $V(\mathcal{G}) =$

$\{1, \ldots, n\}$ denotes the vertex set and $E(\mathcal{G})$ denotes the edge set. Each vertex of this graph may be labeled with one of two states $\{0, 1\}$ and thus a labeling of a graph may be denoted by a vector $\boldsymbol{u} \in \{0, 1\}^n$ where $u_i$ denotes the label of vertex $i$. The *cutsize* of a labeling $\boldsymbol{u}$ is defined as $\phi_{\mathcal{G}}(\boldsymbol{u}) := \sum_{(i,j) \in E(\mathcal{G})} |u_i - u_j|$. The Ising probability distribution over labelings of $\mathcal{G}$ is then defined as $p_{\tau}^{\mathcal{G}}(\boldsymbol{u}) \propto \exp\left(-\frac{1}{\tau} \phi_{\mathcal{G}}(\boldsymbol{u})\right)$ where $\tau > 0$ is the temperature parameter. In our online setting at the beginning of trial $t + 1$ we will have already received an *example sequence*, $\mathcal{S}_t$, of $t$ vertex-label pairs $(i_1, y_1), \ldots, (i_t, y_t)$ where pair $(i, y) \in V(\mathcal{G}) \times \{0, 1\}$. We use $p_{\tau}^{\mathcal{G}}(u_v = y | \mathcal{S}_t) := p_{\tau}^{\mathcal{G}}(u_v = y | u_{i_1} = y_1, \ldots, u_{i_t} = y_t)$ to denote the marginal probability that vertex $v$ has label $y$ given the previously labeled vertices of $\mathcal{S}_t$. For convenience we also define the marginalized *cutsize* $\phi_{\mathcal{G}}(\boldsymbol{u} | \mathcal{S}_t)$ to be equal to $\phi_{\mathcal{G}}(\boldsymbol{u})$ if $u_{i_1} = y_1, \ldots, u_{i_t} = y_t$ and equal to undefined otherwise. Our prediction $\hat{y}_{t+1}$ of vertex $i_{t+1}$ is then the label with maximal marginal probability in the limit of zero temperature, thus

$$\hat{y}_{t+1}^{0\mathrm{I}}(i_{t+1} | \mathcal{S}_t) := \operatorname*{argmax}_{y \in \{0,1\}} \lim_{\tau \to 0} p_{\tau}^{\mathcal{G}}(u_{i_{t+1}} = y | u_{i_1} = y_1, \ldots, u_{i_t} = y_t). \qquad [\texttt{0-Ising}] \quad (1)$$

Note the prediction is undefined if the labels are equally probable. In low temperatures the mass of the marginal is dominated by the labelings consistent with $\mathcal{S}_t$ and the proposed label of vertex $i_{t+1}$ of minimal cut; as we approach zero, $\hat{y}_{t+1}$ is the label consistent with the maximum number of labelings of minimal cut. Thus if $k := \min_{\boldsymbol{u} \in \{0,1\}^n} \phi_{\mathcal{G}}(\boldsymbol{u} | \mathcal{S})$ then we have that

$$\hat{y}^{0\mathrm{I}}(v | \mathcal{S}) = \begin{cases} 0 & |\boldsymbol{u} \in \{0,1\}^n : \phi_{\mathcal{G}}(\boldsymbol{u} | (\mathcal{S}, (v, 0))) = k| > |\boldsymbol{u} \in \{0,1\}^n : \phi_{\mathcal{G}}(\boldsymbol{u} | (\mathcal{S}, (v, 1))) = k| \\ 1 & |\boldsymbol{u} \in \{0,1\}^n : \phi_{\mathcal{G}}(\boldsymbol{u} | (\mathcal{S}, (v, 0))) = k| < |\boldsymbol{u} \in \{0,1\}^n : \phi_{\mathcal{G}}(\boldsymbol{u} | (\mathcal{S}, (v, 1))) = k| \end{cases}.$$

The problem of counting minimum label-consistent cuts was shown to be #P-complete in [10] and further computing $\hat{y}^{0\mathrm{I}}(v | \mathcal{S})$ is also NP-hard (see Appendix G). In Section 2.1 we introduce the Picard-Queyranne graph [9] which captures the combinatorial structure of the set of minimum-cuts. We then use this simplifying structure as a basis to design a heuristic approximation to $\hat{y}^{0\mathrm{I}}(v | \mathcal{S})$ with a mistake bound guarantee.

**Predicting the labelling of a graph in the mistake bound model.** We prove performance guarantees for our method in the mistake bound model introduced by Littlestone [11]. On the graph this model corresponds to the following game. Nature presents a graph $\mathcal{G}$; Nature queries a vertex $i_1 \in V(\mathcal{G}) = \mathbb{N}_n$; the learner predicts the label of the vertex $\hat{y}_1 \in \{0, 1\}$; nature presents a label $y_1$; nature queries a vertex $i_2$; the learner predicts $\hat{y}_2$; and so forth. The learner's goal is to minimize the total number of mistakes $M = |\{t : \hat{y}_t \neq y_t\}|$. If nature is adversarial, the learner will always make a "mistake", but if nature is regular or simple, there is hope that a learner may incur only a few mistakes. Thus, a central goal of online learning is to design algorithms whose total mistakes can be bounded relative to the complexity of nature's labeling. The graph labeling problem has been studied extensively in the online literature. Here we provide a rough discussion of the two main approaches for graph label prediction, and in Section 3.3 we provide a more detailed comparison. The first approach is based on the graph Laplacian [12, 13, 14]; it provides bounds that utilize the additional connectivity of non-tree graphs, which are particularly strong when the graph contains uniformly-labeled clusters of small (resistance) diameter. The drawbacks of this approach are that the bounds are weaker on graphs with large diameter and that the computation times are slower. The second approach is to estimate the original graph with an appropriately selected tree or "path" graph [15, 16, 1, 17]; this leads to faster computation times, and bounds that are better on graphs with large diameters. The algorithm treeOpt [1] is optimal on trees. These algorithms may be extended to non-tree graphs by first selecting a spanning tree uniformly at random [16] and then applying the algorithm to the sampled tree. This randomized approach enables expected mistake bounds which exploit the cluster structure in the graph.

The bounds we prove for the NP-hard 0-Ising prediction and our heuristic are most similar to the "small p" bounds proven for the $p$-seminorm interpolation algorithm [14]. Although these bounds are not strictly comparable, a key strength of our approach is that the new bounds often improve when the graph contains uniformly-labeled clusters of varying diameters. Furthermore, when the graph is a tree we match the optimal bounds of [1]. Finally, the cumulative time required to compute the complete labeling of a graph is quadratic in the size of the graph for our algorithm, while [14] requires the minimization of a non-strongly convex function (on every trial) which is not differentiable when $p \to 1$.

## 2 Preliminaries

An (undirected) graph $\mathcal{G}$ is a pair of sets $(V, E)$ such that $E$ is a set of *unordered* pairs of distinct elements from $V$. We say that $\mathcal{R}$ is a subgraph $\mathcal{R} \subseteq \mathcal{G}$ iff $V(\mathcal{R}) \subseteq V(\mathcal{G})$ and $E(\mathcal{R}) = \{(i, j) : i, j \in V(\mathcal{R}), (i, j) \in E(\mathcal{G})\}$. Given any subgraph $\mathcal{R} \subseteq G$, we define its *boundary* (or inner border) $\partial_0(\mathcal{R})$, its *neighbourhood* (or exterior border) $\partial_e(\mathcal{R})$ respectively as $\partial_0(\mathcal{R}) := \{j : i \notin V(\mathcal{R}), j \in V(\mathcal{R}), (i, j) \in E(\mathcal{G})\}$, and $\partial_e(\mathcal{R}) := \{i : i \notin V(\mathcal{R}), j \in V(\mathcal{R}), (i, j) \in E(\mathcal{G})\}$, and its exterior edge border $\partial_e^E(\mathcal{R}) := \{(i, j) : i \notin V(\mathcal{R}), j \in V(\mathcal{R}), (i, j) \in E(\mathcal{G})\}$. The length of a subgraph $\mathcal{P}$ is denoted by $|\mathcal{P}| := |E(\mathcal{P})|$ and we denote the diameter of a graph by $D(\mathcal{G})$. A pair of vertices $v, w \in V(\mathcal{G})$ are $\kappa$-*connected* if there exist $\kappa$ edge-disjoint paths connecting them. The *connectivity of a graph*, $\kappa(\mathcal{G})$, is the maximal value of $\kappa$ such that every pair of points in $\mathcal{G}$ is $\kappa$-connected. The *atomic number* $\mathcal{N}_\kappa(\mathcal{G})$ of a graph at connectivity level $\kappa$ is the minimum cardinality $c$ of a partition of $\mathcal{G}$ into subgraphs $\{\mathcal{R}_1, \ldots, \mathcal{R}_c\}$ such that $\kappa(\mathcal{R}_i) \geq \kappa$ for all $1 \leq i \leq c$.

Our results also require the use of *directed-, multi-,* and *quotient-* graphs. Every undirected graph also defines a directed graph where each undirected edge $(i, j)$ is represented by directed edges $(i, j)$ and $(j, i)$. An *orientation* of an undirected graph is an assignment of a direction to each edge, turning the initial graph into a directed graph. In a *multi-graph* the edge set is now a multi-set and thus there may be multiple edges between two vertices. A *quotient-graph* $\mathbb{G}$ is defined from a graph $\mathcal{G}$ and a partition of its vertex set $\{V_i\}_{i=1}^N$ so that $V(\mathbb{G}) := \{V_i\}_{i=1}^N$ (we often call these vertices *super-vertices* to emphasize that they are sets) and the multiset $E(\mathbb{G}) := \{(I, J) : I, J \in V(\mathbb{G}), I \neq J, i \in I, j \in J, (i, j) \in E(\mathcal{G})\}$. We commonly construct a quotient-graph $\mathbb{G}$ by "merging" a collection of super-vertices, for example, in Figure 2 from 2a to 2b where 6 and 9 are merged to "6/9" and also the five `merges` that transforms 2c to 2d.

The set of all *label-consistent minimum-cuts* in a graph with respect to an example sequence $\mathcal{S}$ is $\mathcal{U}_\mathcal{G}^*(\mathcal{S}) := \mathrm{argmin}_{\boldsymbol{u} \in \{0,1\}^n} \phi_\mathcal{G}(\boldsymbol{u}|\mathcal{S})$. The minimum is typically non-unique. For example in Figure 2a, the vertex sets $\{v_1, \ldots, v_4\}, \{v_5, \ldots, v_{12}\}$ correspond to one label-consistent minimum-cut and $\{v_1, \ldots, v_5, v_7, v_8\}, \{v_6, v_9 \ldots, v_{12}\}$ to another (the cutsize is 3). The (uncapacitated) *maximum flow* is the number of edge-disjoint paths between a source and target vertex. Thus in Figure 2b between vertex "1" and vertex "6/9" there are at most 3 simultaneously edge-disjoint paths; these are also not unique, as one path must pass through either vertices $\langle v_{11}, v_{12} \rangle$ or vertices $\langle v_{11}, v_{10}, v_{12} \rangle$. Figure 2c illustrates one such flow $\mathcal{F}$ (just the directed edges). For convenience it is natural to view the maximum flow or the label-consistent minimum-cut as being with respect to only two vertices as in Figure 2a transformed to Figure 2b so that $\mathcal{H} \leftarrow \mathrm{merge}(\mathcal{G}, \{v_6, v_9\})$. The "flow" and the "cut" are related by Menger's theorem which states that the minimum-cut with respect to a source and target vertex is equal to the max flow between them. Given a connected graph $\mathcal{H}$ and source and target vertices $s, t$ the Ford-Fulkerson algorithm [18] can find $k$ edge-disjoint paths from $s$ to $t$ in time $O(k|E(\mathcal{H})|)$ where $k$ is the value of the max flow.

### 2.1 The Picard-Queyranne graph

Given a set of labels there may be multiple label-consistent minimum-cuts as well as multiple maximum flows in a graph. The Picard-Queyranne (PQ) graph [9] reduces this multiplicity as far as is possible with respect to the indeterminacy of the maximum flow. The vertices of the PQ-graph are defined as a super-vertex set on a partition of the original graph's vertex set. Two vertices are contained in the same super-vertex iff they have the same label in *every* label-consistent minimum-cut. An edge between two vertices defines an analogous edge between two super-vertices iff that edge is conserved in *every* maximum flow. Furthermore the edges between super-vertices strictly orient the labels in any label-consistent minimum-cut as may be seen in the formal definition that follows.

First we introduce the following useful notations: let $k_{\mathcal{G},\mathcal{S}} := \min\{\phi_\mathcal{G}(\boldsymbol{u}|\mathcal{S}) : \boldsymbol{u} \in \{0, 1\}^n\}$ denote the minimum-cutsize of $\mathcal{G}$ with respect to $\mathcal{S}$; let $i \overset{\mathcal{S}}{\sim} j$ denote an equivalence relation between vertices in $V(\mathcal{G})$ where $i \overset{\mathcal{S}}{\sim} j$ iff $\forall \boldsymbol{u} \in \mathcal{U}_\mathcal{G}^*(\mathcal{S}) : u_i = u_j$; and then we define,

**Definition 1** ([9]). *The* Picard-Queyranne graph $\mathbb{G}(\mathcal{G}, \mathcal{S})$ *is derived from graph* $\mathcal{G}$ *and non-trivial example sequence* $\mathcal{S}$. *The graph is an orientation of the quotient graph derived from the partition* $\{\bot, I_2, \ldots, I_{N-1}, \top\}$ *of* $V(\mathcal{G})$ *induced by* $\overset{\mathcal{S}}{\sim}$. *The edge set of* $\mathbb{G}$ *is constructed of* $k_{\mathcal{G},\mathcal{S}}$ *edge-disjoint paths starting at source vertex* $\bot$ *and terminating at target vertex* $\top$. *A labeling* $\boldsymbol{u} \in \{0, 1\}^n$ *is in* $\mathcal{U}_\mathcal{G}^*(\mathcal{S})$ *iff*

    *1. $i \in \bot$ implies $u_i = 0$ and $i \in \top$ implies $u_i = 1$*

2. $i, j \in H$ *implies* $u_i = u_j$

3. $i \in I, j \in J, (I, J) \in E(\mathbb{G})$*, and* $u_i = 1$ *implies* $u_j = 1$

*where* $\perp$ *and* $\top$ *are the source and target vertices and* $H, I, J \in V(\mathbb{G})$.

As $\mathbb{G}(\mathcal{G}, \mathcal{S})$ is a DAG it naturally defines a partial order $(V(\mathbb{G}), \leq_{\mathbb{G}})$ on the vertex set where $I \leq_{\mathbb{G}} J$ if there exists a path starting at $I$ and ending at $J$. The least and greatest elements of the partial order are $\perp$ and $\top$. The notation $\uparrow R$ and $\downarrow R$ denote the *up set* and *down set* of $R$. Given the set $\mathcal{U}^*$ of all label-consistent minimum-cuts then if $\boldsymbol{u} \in \mathcal{U}^*$ there exists an antichain $A \subseteq V(\mathbb{G}) \setminus \{\top\}$ such that $u_i = 0$ when $i \in I \in \downarrow A$ otherwise $u_i = 1$; furthermore for every antichain there exists a label-consistent minimum-cut. The simple structure of $\mathbb{G}(\mathcal{G}, \mathcal{S})$ was utilized by [9] to enable the efficient algorithmic enumeration of minimum-cuts. However, the cardinality of this set of all label-consistent minimum-cuts is potentially exponential in the size of the PQ-graph and the exact computation of the cardinality was later shown #P-complete in [10]. In Figure 1 we give the algorithm from [9, 19]

**PicardQueyranneGraph**(*graph:* $\mathcal{G}$; *example sequence:* $\mathcal{S} = (v_k, y_k)_{k=1}^t$)

1. $(\mathcal{H}, s, t) \leftarrow \text{SourceTargetMerge}(\mathcal{G}, \mathcal{S})$
2. $\mathcal{F} \leftarrow \text{MaxFlow}(\mathcal{H}, s, t)$
3. $\mathcal{I} \leftarrow (V(\mathcal{I}), E(\mathcal{I}))$ where $V(\mathcal{I}) := V(\mathcal{H})$ and $E(\mathcal{I}) := \{(i, j) : (i, j) \in E(\mathcal{H}), (j, i) \notin \mathcal{F}\}$
4. $\mathbb{G}^0 \leftarrow \text{QuotientGraph}(\text{StronglyConnectedComponents}(\mathcal{I}), \mathcal{H})$
5. $E(\mathbb{G}) \leftarrow E(\mathbb{G}^0); V(\mathbb{G}) \leftarrow V(\mathbb{G}^0)$ except $\perp(\mathbb{G}) \leftarrow \perp(\mathbb{G}^0) \cup \{v_k : k \in \mathbb{N}_t, y_k = 0\}$
   and $\top(\mathbb{G}) \leftarrow \top(\mathbb{G}^0) \cup \{v_k : k \in \mathbb{N}_t, y_k = 1\}$

**Return:** *directed graph:* $\mathbb{G}$

Figure 1: Computing the Picard-Queyranne graph

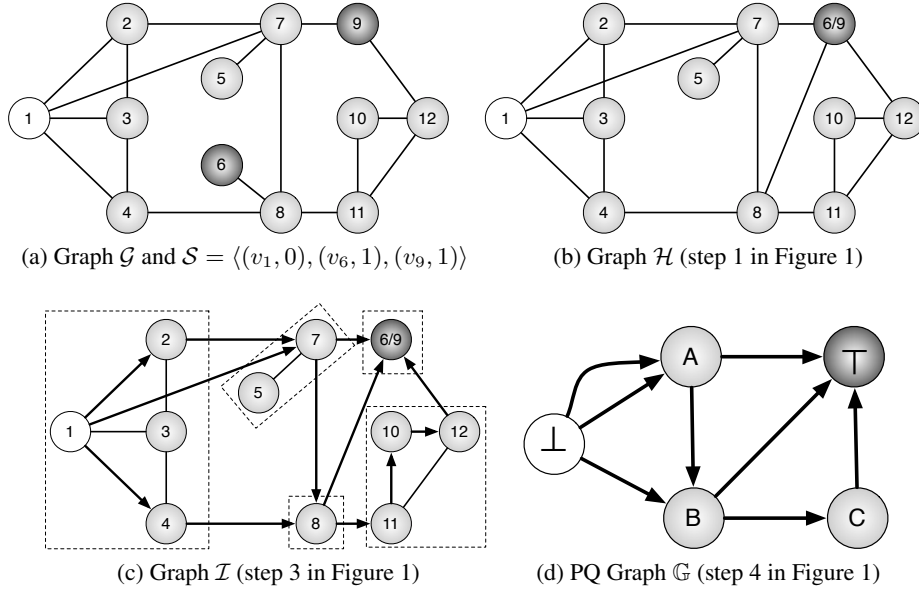

(a) Graph $\mathcal{G}$ and $\mathcal{S} = \langle (v_1, 0), (v_6, 1), (v_9, 1) \rangle$

(b) Graph $\mathcal{H}$ (step 1 in Figure 1)

(c) Graph $\mathcal{I}$ (step 3 in Figure 1)

(d) PQ Graph $\mathbb{G}$ (step 4 in Figure 1)

Figure 2: Building a Picard-Queyranne graph

to compute a PQ-graph. We illustrate the computation in Figure 2. The algorithm operates first on $(\mathcal{G}, \mathcal{S})$ (step 1) by "`merging`" all vertices which share the same label in $\mathcal{S}$ to create $\mathcal{H}$. In step 2 a max flow graph $\mathcal{F} \subseteq \mathcal{H}$ is computed by the Ford-fulkerson algorithm. It is well-known in the case of unweighted graphs that a max flow graph $\mathcal{F}$ may be output as a DAG of $k$ edge-disjoint paths where $k$ is the value of the flow. In step 3 all edges in the flow become directed edges creating $\mathcal{I}$. The graph $\mathbb{G}^0$ is then created in step 4 from $\mathcal{I}$ where the strongly connected components become the super-vertices of $\mathbb{G}^0$ and the super-edges correspond to a subset of flow edges from $\mathcal{F}$. Finally, in

step 5, we create the PQ-graph $\mathbb{G}$ by "fixing" the source and target vertices so that they also have as elements the original labeled vertices from $\mathcal{S}$ which were merged in step 1. The correctness of the algorithm follows from arguments in [9]; we provide an independent proof in Appendix B.

**Theorem 2** ([9]). *The algorithm in Figure 1 computes the unique Picard-Queyranne graph $\mathbb{G}(\mathcal{G}, \mathcal{S})$ derived from graph $\mathcal{G}$ and non-trivial example sequence $\mathcal{S}$.*

## 3 Mistake Bounds Analysis

In this section we analyze the mistakes incurred by the intractable `0-Ising` strategy (see (1)) and the strategy `longest-path` (see Figure 3). Our analysis splits into two parts. Firstly, we show (Section 3.1, Theorem 4) for a sufficiently *regular* graph label prediction algorithm, that we may analyze *independently* the mistake bound of each uniformly-labeled cluster (connected subgraph). Secondly, the per-cluster analysis then separates into three cases, the result of which is summarized in Theorem 10. For a given cluster $\mathcal{C}$ when its internal connectivity is larger than the number of edges in the boundary ($\kappa(\mathcal{C}) > |\partial_e^E(\mathcal{C})|$) we will incur no more than one mistake in that cluster. On the other hand for smaller connectivity clusters ($\kappa(\mathcal{C}) \leq |\partial_e^E(\mathcal{C})|$) we incur up to quadratically in mistakes via the edge boundary size. When $\mathcal{C}$ is a tree we incur $\mathcal{O}(|\partial_e^E(\mathcal{C})| \log D(\mathcal{C}))$ mistakes.

The analysis of smaller connectivity clusters separates into two parts. First, a sequence of trials in which the label-consistent minimum-cut does not increase, we call a *PQ-game* (Section 3.2) as in essence it is played on a PQ-graph. We give a mistake bound for a PQ-game for the intractable `0-Ising` prediction and a comparable bound for the strategy `longest-path` in Theorem 8. Second, when the label-consistent minimum-cut increases the current PQ-game ends and a new one begins, leading to a sequence of PQ-games. The mistakes incurred over a sequence of PQ-games is addressed in the aforementioned Theorem 10 and finally Section 3.3 concludes with a discussion of the combined bounds of Theorems 4 and 10 with respect to other graph label prediction algorithms.

### 3.1 Per-cluster mistake bounds for *regular* graph label prediction algorithms

An algorithm is called *regular* if it is *permutation-invariant*, *label-monotone*, and *Markov*. An algorithm is *permutation-invariant* if the prediction at any time $t$ does not depend on the order of the examples up to time $t$; *label-monotone* if for every example sequence if we insert an example "between" examples $t$ and $t+1$ with label $y$ then the prediction at time $t+1$ is unchanged or changed to $y$; and *Markov* with respect to a graph $\mathcal{G}$ if for any disjoint vertex sets $P$ and $Q$ and separating set $R$ then the predictions in $P$ are independent of the labels in $Q$ given the labels of $R$. A subgraph is *uniformly-labeled* with respect to an example sequence iff the label of each vertex is the same and these labels are consistent with the example sequence. The following definition characterizes the "worst-case" example sequences for regular algorithms with respect to uniformly-labeled clusters.

**Definition 3.** *Given an online algorithm $\mathcal{A}$ and a uniformly-labeled subgraph $\mathcal{C} \subseteq \mathcal{G}$, then $\mathcal{B}_{\mathcal{A}}(\mathcal{C}; \mathcal{G})$ denotes the maximal mistakes made only in $\mathcal{C}$ for the presentation of any permutation of examples in $\partial_e(\mathcal{C})$, each with label $y$, followed by any permutation of examples in $\mathcal{C}$, each with label $1-y$.*

The following theorem enables us to analyze the mistakes incurred in each uniformly-labeled subgraph $\mathcal{C}$ *independently* of each other and *independently* of the remaining graph structure excepting the subgraph's exterior border $\partial_e(\mathcal{C})$.

**Theorem 4** (Proof in Appendix D). *Given an online permutation-invariant label-monotone Markov algorithm $\mathcal{A}$ and a graph $\mathcal{G}$ which is covered by uniformly-labeled subgraphs $\mathcal{C}_1, \ldots, \mathcal{C}_c$ the mistakes incurred by the algorithm may be bounded by $M \leq \sum_{i=1}^{c} \mathcal{B}_{\mathcal{A}}(\mathcal{C}_i; \mathcal{G})$.*

The above theorem paired with Theorem 10 completes the mistake bound analysis of our algorithms.

### 3.2 PQ-games

Given a PQ-graph $\mathbb{G} = \mathbb{G}(\mathcal{G}, \mathcal{S})$, the derived online PQ-game is played between a `player` and an `adversary`. The aim of the player is to minimize their mistaken predictions; for the adversary it is to maximize the player's mistaken predictions. Thus to play the adversary proposes a vertex $z \in Z \in V(\mathbb{G})$, the player then predicts a label $\hat{y} \in \{0, 1\}$, then the adversary returns a label $y \in \{0, 1\}$ and either a *mistake* is incurred or not. The only restriction on the adversary is to not return label which increases the label-consistent minimum-cut. As long as the adversary does not give an example $(z \in \bot, 1)$ or $(z \in \top, 0)$, the label-consistent minimum-cut does not increase

no matter the value of $y$; which also implies the player has a trivial strategy to predict the label of $z \in \bot \cup \top$. After the example is given, we have an updated PQ-graph with new source and target super-vertices as seen in the proposition below.

**Proposition 5.** *If* $\mathbb{G}(\mathcal{G}, \mathcal{S})$ *is a PQ-graph and* $(z, y = 0)$ $((z, y = 1))$ *is an example with* $z \in Z \in V(\mathbb{G})$ *and* $z \notin \top$ $(z \notin \bot)$ *then let* $\mathbb{Z} = \downarrow\{Z\}$ $(\mathbb{Z} = \uparrow\{Z\})$ *then* $\mathbb{G}(\mathcal{G}, \langle \mathcal{S}, (z, y) \rangle) =$ $\mathrm{merge}(\mathbb{G}(\mathcal{G}, \mathcal{S}), \mathbb{Z})$.

Thus given the PQ-graph $\mathbb{G}$ the PQ-game is independent of $\mathcal{G}$ and $\mathcal{S}$, since a "play" $z \in V(\mathcal{G})$ induces a "play" $Z \in V(\mathbb{G})$ (with $z \in Z$).

**Mistake bounds for PQ-games.** Given a *single* PQ-game, in the following we will discuss the three strategies `fixed-paths`, `0-Ising`, and `longest-path` that the `player` may adopt for which we prove online mistake bounds. The first strategy `fixed-paths` is merely motivational: it can be used to play a *single* PQ-game, but not a sequence. The second strategy `0-Ising` is computationally infeasible. Finally, the `longest-path` strategy is "dynamically" similar to `fixed-paths` but is also permutation-invariant. Common to all our analyses is a $k$-*path cover* $P$ of PQ-graph $\mathbb{G}$ which is a partitioning of the edge-set of $\mathbb{G}$ into $k$ edge-disjoint directed paths $P := \{p^1, \ldots, p^k\}$ from $\bot$ to $\top$. Note that the cover is not necessarily unique; for example, in Figure 2d, we have the two unique path covers $P_1 := \{(\bot, A, \top), (\bot, A, B, \top), (\bot, B, C, \top)\}$ and $P_2 := \{(\bot, A, \top), (\bot, A, B, C, \top), (\bot, B, \top)\}$. We denote the set of all path covers as $\mathcal{P}$ and thus we have for Figure 2d that $\mathcal{P} := \{P_1, P_2\}$. This cover motivates a simple mistake bound and strategy. Suppose we had a single path of length $|p|$ where the first and last vertex are the "source" and "target" vertices. So the minimum label-consistent cut-size is "1" and a natural strategy is simply to predict with the "nearest-neighbor" revealed label and trivially our mistake bound is $\log |p|$. Generalizing to multiple paths we have the following strategy.

**Strategy** `fixed-paths`$(\widetilde{P})$**:** Given a PQ-graph choose a path cover $\{\tilde{p}^1, \ldots, \tilde{p}^k\} = \widetilde{P} \in \mathcal{P}(\mathbb{G})$. If the path cover is also vertex-disjoint except for the source and target vertex we may directly use the "nearest-neighbor" strategy detailed above, achieving the mistake upper bound $M \leq \sum_{i=1}^{k} \log |\tilde{p}^i|$. Unsurprisingly, in the vertex-disjoint case it is a mistake-bound optimal [11] algorithm. If, however, $\widetilde{P}$ is not vertex-disjoint and we need to predict a vertex $V$ we may select a path in $\widetilde{P}$ containing $V$ and predict with the nearest neighbour and also obtain the bound above. In this case, however, the bound may not be "optimal." Essentially the same technique was used in [20] in a related setting for learning "directed cuts." A limitation of the `fixed-paths` strategy is that it does not seem possible to extend into a strategy that can play a *sequence* of PQ-games and still meet the regularity properties, particularly permutation-invariance as required by Theorem 4.

**Strategy** `0-Ising`**:** The prediction of the Ising model in the limit of zero temperature (cf. (1)), is equivalent to those of the well-known *Halving* algorithm [21, 22] where the hypothesis class $\mathcal{U}^*$ is the set of label-consistent minimum-cuts. The mistake upper bound of the *Halving algorithm* is just $M \leq \log |\mathcal{U}^*|$ where this bound follows from the observation that whenever a mistake is made at least "half" of concepts in $\mathcal{U}^*$ are no longer consistent. We observe that we may upper bound $|\mathcal{U}^*| \leq \mathrm{argmin}_{P \in \mathcal{P}(\mathbb{G})} \prod_{i=1}^{k} |p^i|$ since the product of path lengths from *any* path cover $P$ is an upper bound on the cardinality of $\mathcal{U}^*$ and hence we have the bound in (2). And in fact this bound may be a significant improvement over the `fixed-paths` strategy's bound as seen in the following proposition.

**Proposition 6** (Proof in Appendix C)**.** *For every* $c \geq 2$ *there exists a PQ-graph* $\mathbb{G}_c$, *with a path cover* $P' \in \mathcal{P}(\mathbb{G}_c)$ *and a PQ-game example sequence such that the mistakes* $M_{\mathtt{fixed-paths}(P')} = \Omega(c^2)$, *while for all PQ-game example sequences on* $\mathbb{G}_c$ *the mistakes* $M_{\mathtt{0-Ising}} = \mathcal{O}(c)$.

Unfortunately the `0-Ising` strategy has the drawback that counting label-consistent minimum-cuts is #P-complete and computing the prediction (see (1)) is NP-hard (see Appendix G).

**Strategy** `longest-path`**:** In our search for an efficient and *regular* prediction strategy it seems natural to attempt to "dynamize" the `fixed-paths` approach and predict with a nearest neighbor along a dynamic path. Two such permutation-invariant methods are the `longest-path` and `shortest-path` strategies. The strategy `shortest-path` predicts the label of a super-vertex $Z$ in a PQ-game $\mathbb{G}$ as 0 iff the shortest directed path $(\bot, \ldots, Z)$ is shorter than the shortest directed path $(Z, \ldots, \top)$. The strategy `longest-path` predicts the label of a super-vertex $Z$ in a PQ-game $\mathbb{G}$ as 0 iff the longest directed path $(\bot, \ldots, Z)$ is shorter than the longest directed path $(Z, \ldots, \top)$. The strategy `shortest-path` seems to be intuitively favored over `longest-path` as it is just

**Input:** Graph: $\mathcal{G}$, Example sequence: $\mathcal{S} = \langle (i_1,0), (i_2,1), (i_3,y_3), \ldots, (i_\ell,y_\ell) \rangle \in (\mathbb{N}_n \times \{0,1\})^\ell$
**Initialization:** $\mathbb{G}_3 = \mathtt{PicardQueyranneGraph}(\mathcal{G}, \mathcal{S}_2)$
**for** $t = 3, \ldots, \ell$ **do**

    **Receive:** $i_t \in \{1, \ldots, n\}$

    $I_t = V \in V(\mathbb{G}_t)$ with $i_t \in V$

    **Predict** ($\mathtt{longest\text{-}path}$): $\hat{y}_t = \begin{cases} 0 & |\mathtt{longest\text{-}path}(\mathbb{G}_t, \bot_t, I_t)| \leq |\mathtt{longest\text{-}path}(\mathbb{G}_t, I_t, \top_t)| \\ 1 & \textbf{otherwise} \end{cases}$

    **Predict** ($\mathtt{0\text{-}Ising}$):       $\hat{y}_t = \hat{y}^{\mathtt{I0}}(i_t | \mathcal{S}_{t-1})$           % as per equation (1)

    **Receive:** $y_t$

    **if** $(i_t \notin \bot_t$ or $y_t \neq 1)$ and $(i_t \notin \top_t$ or $y_t \neq 0)$ **then**    % cut unchanged

       $\mathbb{G}_{t+1} = \begin{cases} \mathtt{merge}(\mathbb{G}_t, \downarrow\{I_t\}) & y_t = 0 \\ \mathtt{merge}(\mathbb{G}_t, \uparrow\{I_t\}) & y_t = 1 \end{cases}$

    **else**                               % cut increases

       $\mathbb{G}_{t+1} = \mathtt{PicardQueyranneGraph}(\mathcal{G}, \mathcal{S}_t)$

**end**

Figure 3: $\mathtt{Longest\text{-}path}$ and $\mathtt{0\text{-}Ising}$ online prediction

the "nearest-neighbor" prediction with respect to the geodesic distance. However, the following proposition shows that it is strictly worse than any $\mathtt{fixed\text{-}paths}$ strategy in the worst case.

**Proposition 7** (Proof in Appendix C). *For every $c \geq 4$ there exists a PQ-graph $\mathbb{G}_c$ and a PQ-game example sequence such that the mistakes $M_{\mathtt{shortest\text{-}path}} = \Omega(c^2 \log(c))$, while for every path cover $P \in \mathcal{P}(\mathbb{G}_c)$ and for all PQ-game example sequences on $\mathbb{G}_c$ the mistakes $M_{\mathtt{fixed\text{-}paths}(P)} = \mathcal{O}(c^2)$.*

In contrast, for the strategy $\mathtt{longest\text{-}paths}$ in the proof of Theorem 8 we show that there always exists some retrospective path cover $P_{\mathtt{lp}} \in \mathcal{P}(\mathbb{G})$ such that $M_{\mathtt{longest\text{-}paths}} \leq \sum_{i=1}^{k} \log |p_{\mathtt{lp}}^i|$. Computing the "longest-path" has time complexity linear in the number of edges in a DAG.

Summarizing the mistake bounds for the three PQ-game strategies for a single PQ-game we have the following theorem.

**Theorem 8** (Proof in Appendix C). *The mistakes, $M$, of an online PQ-game for player strategies* $\mathtt{fixed\text{-}paths}(\widetilde{P})$, $\mathtt{0\text{-}Ising}$, *and* $\mathtt{longest\text{-}path}$ *on PQ-graph $\mathbb{G}$ and k-path cover $\widetilde{P} \in \mathcal{P}(\mathbb{G})$ is bounded by*

$$M \leq \begin{cases} \sum_{i=1}^{k} \log |\tilde{p}^i| & \mathtt{fixed\text{-}paths}(\widetilde{P}) \\ \mathrm{argmin}_{P \in \mathcal{P}(\mathbb{G})} \sum_{i=1}^{k} \log |p^i| & \mathtt{0\text{-}Ising} \\ \mathrm{argmax}_{P \in \mathcal{P}(\mathbb{G})} \sum_{i=1}^{k} \log |p^i| & \mathtt{longest\text{-}path} \end{cases}. \tag{2}$$

### 3.3 Global analysis of prediction at zero temperature

In Figure 3 we summarize the prediction protocol for $\mathtt{0\text{-}Ising}$ and $\mathtt{longest\text{-}path}$. We claim the regularity properties of our strategies in the following theorem.

**Theorem 9** (Proof in Appendix E). *The strategies* $\mathtt{0\text{-}Ising}$ *and* $\mathtt{longest\text{-}path}$ *are permutation-invariant, label-monotone, and Markov.*

The technical hurdle here is to prove that label-monotonicity holds over a sequence of PQ-games. For this we need an analog of Proposition 5 to describe how the PQ-graph changes when the label-consistent minimum-cut increases (see Proposition 19). The application of the following theorem along with Theorem 4 implies we may bound the mistakes of each uniformly-labeled cluster in potentially three ways.

**Theorem 10** (Proof in Appendix D). *Given either the* $\mathtt{0\text{-}Ising}$ *or* $\mathtt{longest\text{-}path}$ *strategy $\mathcal{A}$ the mistakes on uniformly-labeled subgraph $\mathcal{C} \subseteq \mathcal{G}$ are bounded by*

$$\mathcal{B}_{\mathcal{A}}(\mathcal{C}; \mathcal{G}) \in \begin{cases} \mathcal{O}(1) & \kappa(\mathcal{C}) > |\partial_e^E(\mathcal{C})| \\ \mathcal{O}\left(|\partial_e^E(\mathcal{C})|(1 + |\partial_e^E(\mathcal{C})| - \kappa(\mathcal{C})) \log N(\mathcal{C})\right) & \kappa(\mathcal{C}) \leq |\partial_e^E(\mathcal{C})| \\ \mathcal{O}(|\partial_e^E(\mathcal{C})| \log D(\mathcal{C})) & \mathcal{C} \text{ is a tree} \end{cases} \tag{3}$$

*with the atomic number $N(\mathcal{C}) := \mathcal{N}_{|\partial_e^E(\mathcal{C})|+1}(\mathcal{C}) \leq |V(\mathcal{C})|$.*

First, if the internal connectivity of the cluster is high we will only make a single mistake in that cluster. Second, if the cluster is a tree then we pay the external connectivity of the cluster $|\partial_e^E(\mathcal{C})|$ times the log of the cluster diameter. Finally, in the remaining case we pay quadratically in the external connectivity and logarithmically in the "atomic number" of the cluster. The atomic number captures the fact that even a poorly connected cluster may have sub-regions of high internal connectivity.

**Computational complexity.** If $\mathcal{G}$ is a graph and $\mathcal{S}$ an example sequence with a label-consistent minimum-cut of $\phi$ then we may implement the `longest-path` strategy so that it has a cumulative computational complexity of $\mathcal{O}(\max(\phi, n)\,|E(\mathcal{G})|)$. This follows because if on a trial the "cut" does not increase we may implement prediction and update in $\mathcal{O}(|E(\mathcal{G})|)$ time. On the other hand if the "cut" increases by $\phi'$ we pay $\mathcal{O}(\phi'|E(\mathcal{G})|)$ time. To do so we implement an online "Ford-Fulkerson" algorithm [18] which starts from the previous "residual" graph to which it then adds the additional $\phi'$ flow paths with $\phi'$ steps of size $\mathcal{O}(|E(\mathcal{G})|)$.

**Discussion.** There are essentially five dominating mistake bounds for the online graph labeling problem: (I) the bound of `treeOpt` [1] on trees, (II) the bound in expectation of `treeOpt` on a random spanning tree sampled from a graph [1], (III) the bound of `p-seminorm interpolation` [14] tuned for "sparsity" ($p < 2$), (IV) the bound of `p-seminorm interpolation` as tuned to be equivalent to online label propagation [5] ($p = 2$), (V) this paper's `longest-path` strategy.

The algorithm `treeOpt` was shown to be optimal on trees. In Appendix F we show that `longest-path` also obtains the same optimal bound on trees. Algorithm (II) applies to generic graphs and is obtained from (I) by sampling a random spanning tree (RST). It is not directly comparable to the other algorithms as its bound holds only in *expectation* with respect to the RST.

We use [14, Corollary 10] to compare (V) to (III) and (IV). We introduce the following simplifying notation to compare bounds. Let $\mathcal{C}_1, \ldots, \mathcal{C}_c$ denote uniformly-labeled clusters (connected subgraphs) which cover the graph and set $\kappa_r := \kappa(\mathcal{C}_r)$ and $\phi_r := |\partial_e^E(\mathcal{C}_r)|$. We define $D_{r(i)}$ to be the *wide diameter* at connectivity level $i$ of cluster $\mathcal{C}_r$. The wide diameter $D_{r(i)}$ is the minimum value such that for all pairs of vertices $v, w \in \mathcal{C}_r$ there exists $i$ edge-disjoints of paths from $v$ to $w$ of length *at least* $D_{r(i)}$ in $\mathcal{C}_r$ (and if $i > \kappa_r$ then $D_{r(i)} := +\infty$). Thus $D_{r(1)}$ is the diameter of cluster $\mathcal{C}_r$ and $D_{r(1)} \leq D_{r(2)} \leq \cdots$. Let $\phi$ denote the minimum label-consistent cutsize and observe that if the cardinality of the cover $|\{\mathcal{C}_1, \ldots, \mathcal{C}_c\}|$ is minimized then we have that $2\phi = \sum_{r=1}^{c} \phi_r$.

Thus using [14, Corollary 10] we have the following upper bounds of (III): $(\phi/\kappa^*)^2 \log D^* + c$ and (IV): $(\phi/\kappa^*)D^* + c$ where $\kappa^* := \min_r \kappa_r$ and $D^* := \max_r D_r(\kappa^*)$. In comparison we have (V): $\left[\sum_{r=1}^{c} \max(0, \phi_r - \kappa_r + 1)\phi_r \log N_r\right] + c$ with atomic numbers $N_r := \mathcal{N}_{\phi_r+1}(\mathcal{C}_r)$. To contrast the bounds, consider a double lollipop labeled-graph: first create a lollipop which is a path of $n/4$ vertices attached to a clique of $n/4$ vertices. Label these vertices 1. Second, clone the lollipop except with labels 0. Finally join the two cliques with $n/8$ edges arbitrarily. For (III) and (IV) the bounds are $\mathcal{O}(n)$ independent of the choice of clusters. Whereas an upper bound for (V) is the exponentially smaller $\mathcal{O}(\log n)$ which is obtained by choosing a four cluster cover consisting of the two paths and the two cliques. This emphasizes the generic problem of (III) and (IV): parameters $\kappa^*$ and $D^*$ are defined by the worst clusters; whereas (V) is truly a per-cluster bound. We consider the previous "constructed" example to be representative of a generic case where the graph contains clusters of many resistance diameters as well as sparse interconnecting "background" vertices.

On the other hand, there are cases in which (III,IV) improve on (V). For a graph with only small diameter clusters and if the cutsize exceeds the cluster connectivity then (IV) improves on (III,V) given the linear versus quadratic dependence on the cutsize. The log-diameter may be arbitrarily smaller than log-atomic-number ((III) improves on (V)) and also vice-versa. Other subtleties not accounted for in the above comparison include the fact a) the wide diameter is a crude upper bound for resistance diameter (cf. [14, Theorem 1]) and b) the clusters of (III,IV) are not required to be uniformly-labeled. Regarding "a)" replacing "wide" with "resistance" does not change the fact the bound now holds with respect to the worst resistance diameter and the example above is still problematic. Regarding "b)" it is a nice property but we do not know how to exploit this to give an example that significantly improves (III) or (IV) over a slightly more detailed analysis of (V). Finally (III,IV) depend on a correct choice of tunable parameter $p$.

Thus in summary (V) matches the optimal bound of (I) on trees, and can often improve on (III,IV) when a graph is naturally covered by label-consistent clusters of different diameters. However (III,IV) may improve on (V) in a number of cases including when the log-diameter is significantly smaller than log-atomic-number of the clusters.

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
