[Supplementary Material · cold-20-long.pdf]

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

# Technical Appendices

## A  Experiments

In this section we present some preliminary experiments that compare the `longest-path` strategy to `treeOpt` [1] and label propagation [5, 4]. The datasets include the four standardized benchmark datasets `USPS 2 vs 3`, `3 vs 8`, `20 newsgroups` and `ISOLET` as well as a constructed dataset `Stripes`. We used our own implementation of `longest-path` and `labelProp`. For the purposes of computational efficiency we ran our experiments in the "batch mode" rather than "online."

We used the benchmark datasets as follows. With the `USPS` datasets we sampled 500 digits from each class. For `ISOLET` we combined "ISOLET1" to "ISOLET5" giving 3900 in class "0" (letters 'A-M') and 3897 class "1" (letters 'N-Z') examples. While for `20 newsgroups` we combined "comp.*" and "rec.*" creating class "0" with 8124 examples and combining "sci.*" and "talk.*" creating class "1" with 8118 examples. Thus all datasets are nearly balanced between the classes. We then constructed a graph for each dataset by computing a "cost" matrix between all examples (patterns) in the dataset, using the Euclidean metric except on `ISOLET` where we used the "cosine distance." We then constructed both an unweighted minimum spanning tree (MST) and a "3-NN" graph (via the cost matrix) and then "unioned" the edge sets together creating our final graph for each of the datasets. The rationale behind the methodology was based on the common empirical observation that 3-NN graphs are often among the most competitive of the unweighted $k$-NN graphs. We then added a MST to ensure that the final graph was connected. This produced a relatively sparse graph that reduced the computational burden for all methods and reduced variance by avoiding model selection. Although it was beyond the scope of our limited study, it may be the case that constructed graphs with higher connectivity could potentially lead to higher accuracies.

In Figure 4 we report our results. We give the mean accuracy (computed over *all* labels in the graph) and its standard deviation from ten runs. For each "column," and each run of 10, we sampled uniformly $\ell/2$ labels from each class. For the `USPS` datasets we also randomly sampled and built a new graph on each run. Finally on each run as `treeOpt` expects a tree we further sampled a uniform random spanning tree as per [1] from the built graph on each of the 10 runs.

Our obervations are as follows. `LabelProp` performs systematically well across all datasets. `treeOpt` tends to have the weakest performance. Note, however, that `treeOpt` is very computationally efficient and it is natural to run with an ensemble of trees to improve performance; this is discussed and experimentally confirmed in [17]. `Longest-path` is competitive and improves on `labelProp` often. But it has a "failure mode" as seen in the first column for the relatively smaller label sets. We observed that when this occurred we are finding small PQ-graphs corresponding to unbalanced trivial label-consistent minimum-cuts.

We also show results on a constructed dataset to illustrate the potential of the algorithm. `Stripes` is a $60 \times 60$ grid graph with toroidal boundary connectivity. Thus each vertex has four neighbors. The problem corresponds to a simple geometric concept of "stripes." We induce the two classes by alternately "coloring" each of the 6 vertical stripes of $10 \times 60$ vertices. For this dataset the performance of `longest-path` strongly dominates. We provide a visualization of a typical PQ-graph from a `Stripes` in Figure 5.

| | | $\ell = 8$ | $\ell = 16$ | $\ell = 32$ | $\ell = 64$ | $\ell = 128$ |
|---|---|---|---|---|---|---|
| USPS 2 vs. 3 | labelProp | $\mathbf{.980} \pm .010$ | $\mathbf{.982} \pm .008$ | $.984 \pm .005$ | $\mathbf{.988} \pm .003$ | $\mathbf{.991} \pm .002$ |
| | treeOpt | $.814 \pm .055$ | $.885 \pm .032$ | $.891 \pm .032$ | $.956 \pm .013$ | $.959 \pm .013$ |
| | longest-path | $.504 \pm .001$ | $.940 \pm .143$ | $\mathbf{.987} \pm .003$ | $\mathbf{.988} \pm .003$ | $.990 \pm .002$ |

| | | $\ell = 8$ | $\ell = 16$ | $\ell = 32$ | $\ell = 64$ | $\ell = 128$ |
|---|---|---|---|---|---|---|
| USPS 3 vs. 8 | labelProp | $\mathbf{.956} \pm .009$ | $\mathbf{.953} \pm .007$ | $.961 \pm .007$ | $.967 \pm .004$ | $.971 \pm .004$ |
| | treeOpt | $.797 \pm .105$ | $.749 \pm .095$ | $.878 \pm .013$ | $.935 \pm .026$ | $.960 \pm .023$ |
| | longest-path | $.505 \pm .001$ | $.600 \pm .184$ | $\mathbf{.969} \pm .006$ | $\mathbf{.971} \pm .004$ | $\mathbf{.972} \pm .003$ |

| | | $\ell = 32$ | $\ell = 128$ | $\ell = 512$ | $\ell = 1600$ | $\ell = 2048$ |
|---|---|---|---|---|---|---|
| ISOLET | labelProp | $\mathbf{.661} \pm .039$ | $\mathbf{.764} \pm .024$ | $.820 \pm .012$ | $.887 \pm .006$ | $.899 \pm .004$ |
| | treeOpt | $.658 \pm .042$ | $.731 \pm .012$ | $\mathbf{.824} \pm .008$ | $.888 \pm .007$ | $.906 \pm .002$ |
| | longest-path | $.524 \pm .033$ | $.726 \pm .016$ | $.799 \pm .016$ | $\mathbf{.906} \pm .007$ | $\mathbf{.921} \pm .006$ |

| | | $\ell = 800$ | $\ell = 1000$ | $\ell = 2000$ | $\ell = 4000$ | $\ell = 6000$ |
|---|---|---|---|---|---|---|
| Newsgroups | labelProp | $\mathbf{.825} \pm .005$ | $\mathbf{.826} \pm .007$ | $\mathbf{.844} \pm .004$ | $\mathbf{.871} \pm .002$ | $\mathbf{.894} \pm .003$ |
| | treeOpt | $.753 \pm .006$ | $.758 \pm .014$ | $.804 \pm .001$ | $.847 \pm .002$ | $.878 \pm .003$ |
| | longest-path | $.549 \pm .004$ | $.798 \pm .013$ | $.839 \pm .005$ | $.867 \pm .003$ | $.890 \pm .002$ |

| | | $\ell = 250$ | $\ell = 450$ | $\ell = 650$ | $\ell = 850$ | $\ell = 1050$ |
|---|---|---|---|---|---|---|
| Stripes | labelProp | $.915 \pm .013$ | $.948 \pm .005$ | $.962 \pm .004$ | $.972 \pm .004$ | $.978 \pm .005$ |
| | treeOpt | $.817 \pm .012$ | $.879 \pm .017$ | $.909 \pm .006$ | $.928 \pm .003$ | $.936 \pm .006$ |
| | longest-path | $\mathbf{.921} \pm .090$ | $\mathbf{.998} \pm .002$ | $\mathbf{.997} \pm .002$ | $\mathbf{.994} \pm .004$ | $\mathbf{.993} \pm .002$ |

Figure 4: Experiments

Figure 5: A PQ-Graph is generated by a `stripes` run ($\ell = 250$). There are 152 super-vertices, the source vertex ($\bot$) is white, the target ($\top$) is black. The value of the flow is 654. Many of the edges encoded represent multi-edges. Of these there are six types: a green edge with a flow of 231, a brown edge with a flow of 7, an orange with a flow of 5, blue with a flow of 2 and the remaining black edges with a flow of 1.

# B  Properties of the PQ-graph (proof of Theorem 2)

We separate the proof of Theorem 2 into two claims.

**Claim 1.** *The Picard-Queyranne graph $\mathbb{G}(\mathcal{G}, \mathcal{S})$ of Definition 1 is uniquely defined.*

**Claim 2.** *The algorithm in Figure 1 computes the Picard-Queyranne graph $\mathbb{G}(\mathcal{G}, \mathcal{S})$.*

### Proof of Claim 1.

To prove Claim 1 we need to show that $V(\mathbb{G})$ and $E(\mathbb{G})$ are unique. Observe that $V(\mathbb{G})$ is trivially unique as it is the partition on $V(\mathcal{G})$ induced by $\overset{\mathcal{S}}{\sim}$.

To prove $E(\mathbb{G})$ is unique, first recall that $\mathbb{G}$ is an orientation of a quotient graph of $\mathcal{G}$ induced by the partition $(V(\mathbb{G}))$ of $V(\mathcal{G})$; thus up to direction the edges of $\mathbb{G}$ are determined uniquely. Hence, all that is left to prove is that the directions of the edges are uniquely determined. If this is not the case then there exists $\mathbb{G}' = (V(\mathbb{G}), E(\mathbb{G}'))$ and $\mathbb{G}'' = (V(\mathbb{G}), E(\mathbb{G}''))$ satisfying Definition 1 such that there exists $(I, J) \in E(\mathbb{G}')$ and $(J, I) \in E(\mathbb{G}'')$. Then for $i \in I$ and $j \in J$, for all $\boldsymbol{u} \in \mathcal{U}_{\mathcal{G}}^*(\mathcal{S})$ we have $u_i = 1 \Leftrightarrow u_j = 1$ by condition 3 applied to $\mathbb{G}'$ and $\mathbb{G}''$. Thus for arbitrary $i \in I$ and $j \in J$ we have that $u_i = u_j$ which contradicts the fact that $I, J$ are distinct elements in the partition induced by $\overset{\mathcal{S}}{\sim}$. ∎

### Proof of Claim 2.

In this proof we define $\mathcal{G}, \mathcal{S}, \mathcal{H}, \mathcal{I}, \mathbb{G}^0$ and $\mathbb{G}$ as in the algorithm of Figure 1. We also define $k := k_{\mathcal{G}, \mathcal{S}}$. In step 2 of the algorithm the graph $\mathcal{H}$ is formed by merging the vertices of $\mathcal{G}$ in the example sequence $\mathcal{S}$ which are labeled 0 and 1 to create source vertex $s$ and target vertex $t$, respectively. In steps 2-4 we then compute the PQ-graph $\mathbb{G}^0(\mathcal{H}, ((s, 0), (t, 1)))$ and then in step 5, we construct $\mathbb{G}(\mathcal{G}, \mathcal{S})$ by "de-merging" vertices $s$ and $t$. Note that the set of all labelings of $\mathcal{H}$ which satisfy $u_s = 0$ and $u_t = 1$ with minimum-cut is isomorphic to the set of all labelings $\mathcal{U}_{\mathcal{G}}^*(\mathcal{S})$ of $\mathcal{G}$ consistent with $\mathcal{S}$ with minimum-cut; and likewise the edge sets $E(\mathcal{H})$ and $E(\mathcal{G})$ are isomorphic. Hence for simplicity in presentation we lightly abuse notation in the following proof at times by identifying $\mathcal{H}$ with $\mathcal{G}$ and $\mathbb{G}^0$ with $\mathbb{G}$.

**Lemma 11.** *The edge set $E(\mathbb{G})$ computed by the algorithm consists of $k$ edge-disjoint paths from $\perp$ to $\top$*

*Proof.* We have, from the Ford-Fulkerson algorithm on an unweighted graph, that the flow $\mathcal{F}$ consists of $k$ edge-disjoint directed paths $F_1, F_2, \ldots, F_k$ from $s$ to $t$. Take path $F_1$ and write it as $(s = f_0, f_1, \ldots, f_m = t)$. For every $i < m$ we have $(f_i, f_{i+1}) \in \mathcal{F}$ so $(f_{i+1}, f_i) \notin \mathcal{F}$ and hence $(f_i, f_{i+1}) \in \mathcal{I}$ so $F_1$ is a directed path in $\mathcal{I}$. For each strongly connected component $H \in V(\mathbb{G})$, let $f^H = \{i \in \mathbb{N}_m : f_i \in H\}$. Suppose we have $i, j, l \in \mathbb{N}_m$ with $i < j < l$ and $i, l \in f^H$ for some $H \in V(\mathbb{G})$. Then since $f_i$ and $f_l$ are in the same strongly connected component (of $\mathcal{I}$) there exists a directed path $p$ in $\mathcal{I}$ from $f_l$ to $f_i$. Hence we have a path $(f_i, f_{i+1}, \ldots, f_j)$ from $f_i$ to $f_j$ and a path $\langle (f_j, f_{j+1}, \ldots, f_l), p \rangle$ from $f_j$ to $f_i$, in $\mathcal{I}$. This implies that $f_i$ and $f_j$ are in the same strongly connected component and hence $f_j \in H$ so $j \in f^H$. We can hence write $f^H = \{c : a \leq c < b\}$ for some $a, b \in \mathbb{N}_{m+1}$ with $a \leq b$. This means we can write $F_1$ as $\langle g^{H_0}, g^{H_1}, \ldots, g^{H_l} \rangle$ for some $l$ and distinct $H_i$ where $g^{H_i}$ is a sequence containing exactly the elements of $\{f_a : a \in f^{H_i}\}$. Since $f_0 = s$ (resp. $f_m = t$), and $f_0 \in f^{H_0}$ (resp. $f_m \in f^{H_l}$) we must have $H_0 = \perp$ (resp. $H_l = \top$). Upon the collapse of $\mathcal{I}$ to $\mathbb{G}$, $F_1$ hence becomes the path $(\perp = H_0, H_1, \ldots, H_l = \top)$, i.e., $F_1$ collapses to a path in $\mathbb{G}$ from $\perp$ to $\top$. This happens for each $F_i$, giving us $k$ edge-disjoint paths in $\mathbb{G}$ from $\perp$ to $\top$. The result is then seen by noting that every directed edge in $\mathbb{G}$ comes from a directed edge in $\mathcal{F}$ (otherwise the edge would be bidirected in $\mathcal{I}$ implying that both vertices on the edge would be in the same strongly connected component (and hence such an edge would disappear)). □

**Lemma 12.** *Every labelling $\boldsymbol{u} \in \mathcal{U}_{\mathcal{G}}^*(\mathcal{S})$ satisfies the conditions in Definition 1 with respect to $\mathbb{G}$ as computed by the algorithm.*

*Proof.* Let $\boldsymbol{u}$ be a labelling in $\mathcal{U}_{\mathcal{G}}^*(\mathcal{S})$. First we prove the useful fact that,

$$\text{if } u_i \neq u_j \text{ and } (i, j) \in E(\mathcal{I}) \text{ then } u_i = 0, u_j = 1 \tag{4}$$

The above is seen since $k$ edges in $\mathcal{F}$ are cut under $\boldsymbol{u}$ (as $\mathcal{F}$ consists of $k$ edge-disjoint directed path graphs from "$s$" to "$t$" with $u_s = 0$ and $u_t = 1$). Thus if there was a cut edge in $E(\mathcal{G}) \setminus E(\mathcal{F})$ the cut of $\boldsymbol{u}$ would be larger than $k$ which is a contradiction. Hence we have (since $u_i \neq u_j$ and $(i,j) \in E(\mathcal{H})$ (as $(i,j) \in E(\mathcal{I})$)) that either $(i,j) \in \mathcal{F}$ or $(j,i) \in \mathcal{F}$. But if $(j,i) \in \mathcal{F}$ then $(i,j) \notin E(\mathcal{I})$ which is a contradiction. Hence we have that $(i,j) \in \mathcal{F}$ so, since $\boldsymbol{u}$ minimises the cut and $u_i \neq u_j$, we must have $u_i = 0$ and $u_j = 1$.

We now prove that $\boldsymbol{u}$ satisfies conditions 1-3 of Definition 1.

*Condition 2:* Suppose, for contradiction, that $i$ and $j$ are in the same super-vertex $H \in V(\mathbb{G})$ and $u_i \neq u_j$. Then without loss of generality assume $u_i = 1$ and $u_j = 0$. Since $i$ and $j$ are in the same strongly connected component of $\mathcal{I}$ there exists a directed path $(i = v_0, v_1, v_2, \ldots, v_{m-1}, v_m = j)$ in $\mathcal{I}$. Since $u_i \neq u_j$ there exists $a < m$ such that $u_{v_a} \neq u_{v_{a+1}}$ so let $b$ be the minimum such $a$. Since $u_{v_l} = u_{v_{l+1}}$ for all $l < b$, we have $u_{v_b} = u_{v_0} = u_i = 1$. But, since $(v_b, v_{b+1}) \in E(\mathcal{I})$, and $u_{v_b} \neq u_{v_{b+1}}$, this contradicts (4).

*Condition 1:* We have a vertex $s \in \bot$ with $u_s = 0$ (since $\boldsymbol{u}$ is label-consistent). Hence, if $i \in \bot$ then, by condition 2 (with $H := \bot$) $u_i = u_s = 0$. The same goes for $\top$ (with 1 instead of 0 and $t$ instead of $s$).

*Condition 3:* Since $(I, J) \in E(\mathbb{G})$ with $I \neq J$ there exists $i' \in I$ and $j' \in J$ such that there exists an edge $(i', j')$ in $\mathcal{I}$. Hence, if $u_i = 1$ then by condition 2 (with $H := I$), $u_{i'} = 1$ which implies, by (4), that $u_{j'} = 1$ and hence, by condition 2 (with $H := J$), $u_j = 1$. $\qquad\square$

We now prove the converse of Lemma 12.

**Lemma 13.** *If a labelling $\boldsymbol{u}$ satisfies the conditions in Definition 1 with respect to $\mathbb{G}$ as computed by the algorithm then we have $\boldsymbol{u} \in \mathcal{U}_{\mathcal{G}}^*(\mathcal{S})$.*

*Proof.* By condition 1, $u_s = 0$ and $u_t = 1$ so $\boldsymbol{u}$ is label-consistent. From the proof of Lemma 11 we have that $\mathbb{G}$ is formed of $k$ edge-disjoint paths $P_1, P_2, P_3, \ldots, P_k$ which are the flow paths $F_1, F_2, F_3, \ldots, F_k$ after collapse. Let's consider $P_1$ and $F_1$. Let $P_1 = (\bot = H_0, H_1, \ldots, H_l = \top)$ and $F_1 = \langle g^{H_0}, g^{H_1}, \ldots, g^{H_l} \rangle$ for $g^{H_i}$ defined as in the proof of Lemma 11. By condition 2 we have that, for any $a \in \mathbb{N}_l$ there exists $u^{H_a} \in \{0,1\}$ such that for all $i \in H_a$, $u_i = u^{H_a}$. Let $b = \min\{a \in \mathbb{N}_l : u^{H_a} = 1\}$ (note that $b$ exists since $u^{H_l} = 1$ (by condition 1 and since $H_l = \top$) and that $b > 0$ since $u^{H_0} = 0$ (by condition 1 and since $H_0 = \bot$)). Suppose, for contradiction, that $u^{H_c} = 0$ for some $c > b$. Then $d = \min\{a > b : u^{H_a} = 0\}$ is defined. Then we have $u^{H_{d-1}} = 1$ so we have a directed edge $(H_{d-1}, H_d)$ in $\mathbb{G}$ with $u^{H_{d-1}} = 1$ and $u^{H_d} = 0$ which contradicts condition 3. We hence have that $H_a = 1$ for all $a \geq b$. By definition of $b$ we also have that $u^{H_a} = 0$ for all $a < b$. Hence, we have that all elements $i$ of the sequence $\langle g^{H_0}, g^{H_1}, \ldots, g^{H_{b-1}} \rangle$ satisfy $u_i = 0$ and all elements $i$ of the sequence $\langle g^{H_b}, g^{H_{b+1}}, \ldots, g^{H_l} \rangle$ satisfy $u_i = 1$. This means that $F_1$ has exactly one cut edge (the edge from the final vertex of $g^{H_{b-1}}$ to the first vertex of $g^{H_b}$). The same argument for all of the $k$ edge-disjoint paths $P_1, P_2, ..P_k$ implies that every flow path has exactly one cut edge which gives us exactly $k$ cut edges in $\mathcal{F}$. Suppose now that we have an edge $(i,j)$ of $\mathcal{G}$ such that $(i,j),(j,i) \notin \mathcal{F}$. Then $(i,j)$ (resp. $(j,i)$) is a path from $i$ to $j$ (resp. $j$ to $i$) in $\mathcal{I}$. This implies that $i$ and $j$ are in the same strongly connected component of $\mathcal{I}$ and hence by condition 2, $u_i = u_j$. This means that all the edges that aren't on flow paths are not cut and hence that there are exactly $k$ cut edges in $\mathcal{G}$. $\qquad\square$

**Lemma 14.** *If $\mathbb{G}$ is the graph produced by the algorithm then the vertex set $V(\mathbb{G})$ is the partition induced by $\overset{\mathcal{S}}{\sim}$ on $V(\mathcal{G})$.*

*Proof.* Since from Lemma 12 we have shown for $V(\mathbb{G})$ as computed by the algorithm the condition

$$\forall \boldsymbol{u} \in \mathcal{U}_{\mathcal{G}}^*(\mathcal{S}) : i, j \in H \in V(\mathbb{G}) \Rightarrow u_i = u_j \,,$$

we now need to show,

$$\forall i, j : i \in H \in V(\mathbb{G}), j \notin H \Rightarrow \exists \boldsymbol{u} \in \mathcal{U}_{\mathcal{G}}^*(\mathcal{S}) : u_i \neq u_j \,.$$

Thus given $i \in H \in V(\mathbb{G})$, $j \in H' \in V(\mathbb{G})$ and $H \neq H'$ we now show there exists a labeling $\boldsymbol{u} \in \mathcal{U}_{\mathcal{G}}^*(\mathcal{S})$ such that $u_i \neq u_j$.

Assume $H' \not\leq_{\mathbb{G}} H$ (else swap $H$ and $H'$). Let $\mathbb{D} := \downarrow\{H\}$ then for all $i \in \bigcup \mathbb{D}$ set $u_i := 0$ and for all $i \in \bigcup(V(\mathbb{G}) \setminus \mathbb{D})$ set $u_i := 1$. We have (since $H \in \mathbb{D}$) that $\boldsymbol{u}$ labels all vertices in $H$ as 0 and (since $H' \notin \mathbb{D}$) that $\boldsymbol{u}$ labels all vertices in $H'$ as 1. Hence $\boldsymbol{u}$ labels $H$ and $H'$ differently. Hence, all that is required to prove now is that $\boldsymbol{u} \in \mathcal{U}_{\mathcal{G}}^*(\mathcal{S})$ which, by Lemma 13, is proved by showing that $\boldsymbol{u}$ satisfies the conditions 1-3 in Definition 1 which we now show.

*Condition 1:* We have that $\bot \in \mathbb{D}$ and hence all vertices $i \in \bot$ are labelled 0 by $\boldsymbol{u}$. We also have that $\top \notin \mathbb{D}$ (as $H' \leq_{\mathbb{G}} \top$ implies $H \neq \top$) and hence all vertices in $\top$ are labelled 1 by $\boldsymbol{u}$.

*Condition 2:* It is clear from the definition of $\boldsymbol{u}$ that, given $H \in V(\mathbb{G})$, all vertices $i$ in $H$ have the same label $u_i$ since if $H \in \mathbb{D}$ then all vertices in $H$ are labelled 0 and if $H \notin \mathbb{D}$ then all vertices in $H$ are labelled 1.

*Condition 3:* Suppose we have $(I, J) \in E(\mathbb{G})$ and $i \in I$ with $u_i = 1$. We now prove that for all $j \in J$, $u_j = 1$ which shows that $\boldsymbol{u}$ satisfies condition 3. To prove this assume the converse: that $u_j = 0$. Then we must have that $J \in \mathbb{D}$ so there exists a path $P$ in $\mathbb{G}$ from $J$ to $H$. Hence we have a path $\langle(I, J), P\rangle$ from $I$ to $H$ in $\mathbb{G}$ which implies that $I \in \mathbb{D}$ and hence all vertices in $I$, and hence $i$, are labelled 0 by $\boldsymbol{u}$ which is a contradiction. $\square$

Lemmas 11-14 show that $\mathbb{G}$ satisfies Claim 2. $\blacksquare$

## C  PQ-game proofs (Propositions 5,6,7 and Theorem 8)

### C.1  Proof of Proposition 5

*Proof.* By symmetry we can, without loss of generality, assume that $y = 1$. Let $\bot'$ and $\top'$ be the source and target vertices of $\mathbb{G}(\mathcal{G}, \langle \mathcal{S}, (z, y)\rangle)$. Given a labelling $\boldsymbol{u} \in \mathcal{U}_{\mathcal{G}}^*(\mathcal{S})$, define $\boldsymbol{u}'$ to be the labelling of $\mathcal{G}$ such that for all $v \in \bigcup \uparrow\{Z\}$ we have $u_v' := 1$ and for all $v \notin \bigcup \uparrow\{Z\}$ we have $u_v' := u_v$. We now show that given any labelling $\boldsymbol{u} \in \mathcal{U}_{\mathcal{G}}^*(\mathcal{S})$ we have that the cutsize of $\boldsymbol{u}'$ is equal to $k_{\mathcal{G},\mathcal{S}}$ and hence (since $u_z' = 1$) that $\boldsymbol{u}' \in \mathcal{U}_{\mathcal{G}}^*(\langle \mathcal{S}, (z, 1)\rangle)$. To show this split $\mathbb{G}(\mathcal{G}, \mathcal{S})$ into $k_{\mathcal{G},\mathcal{S}}$ edge-disjoint directed paths $p_1, p_2, ..., p_{k_{\mathcal{G},\mathcal{S}}}$. Note that, since $\boldsymbol{u}$ has a cutsize of $k_{\mathcal{G},\mathcal{S}}$, each path $p_i$ has a single cut under $\boldsymbol{u}$. We can write $p_i$ as $(\bot, Y_1, Y_2, ..., Y_m, X_1, X_2, ..., X_m' = \top)$ where each $Y_j$ is not in $\uparrow\{Z\}$ and each $X_j$ is in $\uparrow\{Z\}$. Hence, it is easy to see that since $p_i$ has a single cut under $\boldsymbol{u}$ it also has a single cut under $\boldsymbol{u}'$. $\boldsymbol{u}'$ hence induces a cutsize of $k_{\mathcal{G},\mathcal{S}}$ in $\mathbb{G}(\mathcal{G}, \mathcal{S})$ so $\boldsymbol{u}'$ has a cutsize of $k_{\mathcal{G},\mathcal{S}}$. Note that since (by the above) $k_{\mathcal{G},\langle \mathcal{S},(z,y)\rangle} = k_{\mathcal{G},\mathcal{S}}$ we have $\mathcal{U}_{\mathcal{G}}^*(\langle \mathcal{S}, (z, 1)\rangle) = \{\boldsymbol{u} \in \mathcal{U}_{\mathcal{G}}^*(\mathcal{S}) : u_z = 1\}$.

Given $X \in V(\mathbb{G}(\mathcal{G}, \mathcal{S}))$ and $x, y \in X$ we have, for all $\boldsymbol{u} \in \mathcal{U}_{\mathcal{G}}^*(\langle \mathcal{S}, (z, 1)\rangle)$, $\boldsymbol{u} \in \mathcal{U}_{\mathcal{G}}^*(\mathcal{S})$ and hence, by Item 1 of Definition 1 we have $u_x = u_y$. This implies that $x$ and $y$ are in the same super-vertex of $\mathbb{G}(\mathcal{G}, \langle \mathcal{S}, (z, y)\rangle)$. Hence, given $X \in V(\mathbb{G}(\mathcal{G}, \mathcal{S}))$, we have that $X$ is a subset of some super-vertex, in $\mathbb{G}(\mathcal{G}, \langle \mathcal{S}, (z, y)\rangle)$. Given $X \in V(\mathbb{G}(\mathcal{G}, \mathcal{S}))$ define $\bar{X}$ to be the super-vertex in $\mathbb{G}(\mathcal{G}, \langle \mathcal{S}, (z, y)\rangle)$ that contains $X$ as a subset.

We now show that for every $X \in \uparrow\{Z\}$ we have $\bar{X} = \top'$. Suppose we have $X \in \uparrow\{Z\}$. Then let $(Z = Y_0, Y_1, ..., Y_m = X)$ be a directed path in $\mathbb{G}(\mathcal{G}, \mathcal{S})$. For all $i$ choose $x_i$ to be an arbitrary vertex in $Y_i$. Let $\boldsymbol{u}$ be an arbitrary labelling in $\mathcal{U}_{\mathcal{G}}^*(\langle \mathcal{S}, (z, 1)\rangle)$. Then since $x_0 \in Z$ we have $u_{x_0} = 1$ and hence, by induction on $i$ using Item 3 of Definition 1 (and noting that $\boldsymbol{u} \in \mathcal{U}_{\mathcal{G}}^*(\mathcal{S})$) we have $u_{x_i} = 1$. This implies that $u_{x_m} = 1$. We have just shown that for every labelling $\boldsymbol{u} \in \mathcal{U}_{\mathcal{G}}^*(\langle \mathcal{S}, (z, 1)\rangle)$, $\boldsymbol{u}$ labels all vertices in $X$ as 1 and hence we have that $X \subseteq \top'$ which implies that $\bar{X} = \top'$.

We now show that for every $X \in V(\mathbb{G}(\mathcal{G}, \mathcal{S})) \setminus \uparrow\{Z\}$ we have $\bar{X} \neq \top'$. To see this let $x$ be an arbitrary member of $X$. Then since $X \neq \top$ choose a labelling $\boldsymbol{u} \in \mathcal{U}_{\mathcal{G}}^*(\mathcal{S})$ such that $u_x = 0$. We then have $u_x' = 0$ so since (by the above) $\boldsymbol{u}' \in \mathcal{U}_{\mathcal{G}}^*(\langle \mathcal{S}, (z, 1)\rangle)$ we have $x \notin \top'$. Hence $X \not\subseteq \top'$ so we have $\bar{X} \neq \top'$.

We now show that for every $X, Y \in V(\mathbb{G}(\mathcal{G}, \mathcal{S})) \setminus \uparrow\{Z\}$ we have that $\bar{X} \neq \bar{Y}$. To see this choose $x \in X$ and $y \in Y$. Since $x$ and $y$ are in different super-vertices in $\mathbb{G}(\mathcal{G}, \mathcal{S})$ we can choose $\boldsymbol{u} \in \mathcal{U}_{\mathcal{G}}^*(\mathcal{S})$ such that $u_x \neq u_y$. We then have $u_x' = u_x \neq u_y = u_y'$ so, since (by the above) $\boldsymbol{u}' \in \mathcal{U}_{\mathcal{G}}^*(\langle \mathcal{S}, (z, 1)\rangle)$, we have that $x$ and $y$ are in different super-vertices of $\mathbb{G}(\mathcal{G}, \langle \mathcal{S}, (z, y)\rangle)$. It follows that $\bar{X} \neq \bar{Y}$.

Combining the above results we get the following: for every $X \in \uparrow\{Z\}$ we have $\bar{X} = \top'$ and for every $X \in V(\mathbb{G}(\mathcal{G}, \mathcal{S})) \setminus \uparrow\{Z\}$ we have $\bar{X} = X$. By Item 3 of Definition 1 (and since $\mathcal{U}_{\mathcal{G}}^*(\langle \mathcal{S}, (z, 1) \rangle) \subseteq \mathcal{U}_{\mathcal{G}}^*(\mathcal{S})$) we have that the directions of the edges in $\mathbb{G}(\mathcal{G}, \langle \mathcal{S}, (z, y) \rangle)$ are inherited from $\mathbb{G}(\mathcal{G}, \mathcal{S})$. This completes the proof.

$\square$

## C.2 Proof of Propositions 6 and 7

*Proof of Proposition 6.* The comb PQ-graph $\mathbb{G}_{c,r}$ is generated from the labeled graph $\mathcal{G}$ with $cr + 2c - 1$ vertices, and $2c$ labels. We initially have a path,

$$p^0 := (s_0, v_0, v_1, \ldots, v_{cr-1}, v_{cr}, t_0), ,$$

of $cr + 3$ vertices and then every $r$ vertices (from $v_r$ to $v_{(c-1)r}$) we have a bottom vertex $s_i$ and a top vertex $t_i$ forming a three-vertex path $p^i := (s_i, v_{ir}, t_i)$ for $i \in \mathbb{N}_{c-1}$ that intersects $p^0$. Thus if $\{s_0, \ldots, s_{c-1}\}$ are each labeled "0" and $\{t_0, \ldots, t_{c-1}\}$ are each labeled "1"; then the PQ-graph $\mathbb{G}_{c,r}$ will have $cr + 3$ vertices with all source "s"-vertices "glued" together to form $\bot$ and likewise the "t"-vertices form $\top$ (Notationally it is still convenient to refer to these "s" and "t" vertices within "$\bot$" and "$\top$" as they now correspond to edges leaving the super-vertex). Thus $P := \{p^0, \ldots, p^{c-1}\}$ is one path cover (see Figure 6a). However, we also have another "zig-zag" path cover (see Figure 6b)

$$P' = \{(s_0, \ldots, t_1), (s_1, \ldots, t_2), \ldots, (s_{c-2}, \ldots, t_{c-1}), (s_{c-1}, \ldots, t_0)\} .$$

Observe that path cover $P$ has one path of length $cr + 2$ and $c - 1$ paths of length 2 whereas $P'$ has $c$ paths of length $r + 2$; thus the path lengths are very unbalanced in $P$ as opposed to $P'$ which leads to the following mistake bounds $M_{\texttt{0-Ising}} \leq O(\log r + \log c + (c - 1))$ and $M_{\texttt{fixed-paths}(P')} \leq O(c \log r)$.

(a) Path cover $P$ of $\mathbb{G}_{c,r}$

(b) Path cover $P'$ of $\mathbb{G}_{c,r}$

Figure 6: Two path covers of the comb PQ-graph

Now if we assume the true labeling of $\mathcal{G}$ is all "0" except at the "t"-vertices then an adversary may also force $\texttt{fixed-paths}(P')$ to incur $\Omega(c \log r)$ mistakes by forcing $\Omega(\log r)$ mistakes in first path $(s_0, \ldots, t_1)$ (e.g. by first selecting $v_{r/2+1}$ then $v_{3r/4+1}$ then $v_{7r/8+1}$ and so on) and then $\Omega(\log r)$ mistakes in the second path $(s_1, \ldots, t_2)$ and so on. Thus comparing the two algorithms by setting $r := 2^c$ we have that $M_{\texttt{0-Ising}} \leq O(c)$ and $M_{\texttt{fixed-paths}(P')} \geq \Omega(c^2)$. $\square$

*Proof of Proposition 7.* We consider a generalization of the $(c, r)$-comb to the $(b, c, r)$-grid, with $b, c < r$. The $(b, c, r)$-grid is essentially is a "stacking" of $b + 1$ of the $(c, r)$-combs. More precisely we have $b + 1$ directed path graphs of the form

$$p^{i,\cdot} := (s_{i,\cdot}, v_{i,0}, v_{i,1}, \ldots, v_{i,cr-1}, v_{i,cr}, t_{i,\cdot})$$

for $i \in \{0, 1, \ldots, b\}$ each with $cr + 3$ vertices. We then create a grid intersecting these $b + 1$ paths with $c - 1$ paths of the form

$$p^{\cdot \cdot j} := (s_{\cdot, j}, v_{1, jr}, v_{2, jr}, \ldots, v_{b, jr}, t_{\cdot, j})$$

for $j \in \mathbb{N}_{c-1}$, so that the grid has $(b+1)(cr+3) + 2(c-1)$ vertices and a label-consistent minimum-cut size of $b + c$. We assume the true labeling of each vertex is "0" except the "t" vertices. For the fol-

Figure 7: The grid PQ-graph

lowing we define the path segments $z(i, j) := (v_{i, jr}, v_{i, jr+1}, \ldots, v_{i, (j+1)r-1})$ for $i \in \{0, 1, \ldots, b\}$ and $j \in \{0, 1, \ldots, c-1\}$. We now describe an adversarial strategy for the shortest path heuristic: First force, $\Omega(\log(r - b))$ mistakes on $z(\frac{b}{2} + 1, 0)$ by first selecting $v_{\frac{b}{2}+1, \frac{r+b}{2}+1}$, then selecting $v_{\frac{b}{2}+1, \frac{3(r+b)}{4}+1}$, then selecting $v_{\frac{b}{2}+1, \frac{7(r+b)}{8}+1}$ and so on. In the same way then force $\Omega(\log(r - b))$ mistakes on $z(\frac{b}{2} + 1, j)$ for every $1 \leq j \leq c - 1$ in turn. We have now forced $\Omega(c \log(r - b))$ mistakes on $p^{\frac{b}{2}+1, \cdot \cdot}$. In the same way then force $\Omega(c \log(r - b))$ mistakes on path $p^{\frac{3b}{4}+1, \cdot \cdot}$, then $\Omega(c \log(r - b))$ mistakes on path $p^{\frac{7b}{8}+1, \cdot \cdot}$ and so on. This gives us a total of $\Omega(c \log(b) \log(r - b))$ mistakes. Thus, if we set $b = c, r = 2^c$ we have that $M_{\texttt{shortest-path}} = \Omega(c \log(c) \log(2^c - c))$. Noting that $\Omega(\log(2^c - c)) = \Omega(c)$ we hence have that $M_{\texttt{shortest-path}} = \Omega(c^2 \log(c))$.

On the other hand, since there are $O(bcr)$ vertices in the grid and the grid has $b + c$ edge disjoint paths from source to sink we must have that, for any path cover $P$, $M_{\texttt{fixed-paths}(P)} = O((b + c) \log bcr)$. Thus if we set $b = c, r = 2^c$ we have that $M_{\texttt{fixed-paths}(P)} = O(c^2)$. Hence we have $M_{\texttt{shortest-path}} = \Omega(c^2 \log(c))$ and $M_{\texttt{fixed-paths}(P)} = O(c^2)$. $\qquad \square$

## C.3  Proof of Theorem 8

*Proof of Theorem 8.* The bounds for strategies `fixed-paths` and `0-Ising` are straightforward (see discussion in Section 3.2) and we focus on the proof of (2) for the `longest-path` strategy.

Let $(\mathbb{G}_1, \ldots, \mathbb{G}_T)$ be the sequence of PQ-graphs in a PQ-game of cutsize $k$ (where $\mathbb{G}_T$ is the final PQ-graph of cutsize $k$ and hence the mistake made in this graph is not counted towards the mistakes of the PQ-game). Recall that, for any $t \leq T$, the edge set $E(\mathbb{G}_t)$ may be partitioned (non-uniquely) into $k$ edge-disjoint directed paths. For all $t \leq T$ we will constuct $k$ edge-disjoint directed paths $\{p_t^1, p_t^2, \ldots, p_t^k\}$. Note that we are treating each path as a set of edges rather than vertices so that $|p_t^i|$ is the length of the $i$th path in $\mathbb{G}_t$. Let $M_t$ denote the number of mistakes ("backwards cumulative") of `longest-path` made in the sequence $(\mathbb{G}_t, \mathbb{G}_{t+1}, \ldots, \mathbb{G}_T)$. We prove by a "reverse" induction (i.e., from $t = T$ to $t = 1$) that for all $t$ there exist $k$ edge-disjoint directed paths $\{p_t^1, p_t^2, \ldots, p_t^k\}$ from $\bot$ to $\top$ in $\mathbb{G}_t$, and numbers ("mistakes on a path") $\{M_t^1, M_t^2, \ldots, M_t^k\}$ with $M_t^i \leq \log |p_t^i|$ for $1 \leq i \leq k$ such that the "backwards cumulative mistakes" is $M_t = \sum_{i=1}^k M_t^i$.

We now consider the base case of our induction: for $t = T$ we arbitrarily choose $\{p_T^1, p_T^2, \ldots, p_T^k\}$ to be an arbitrary set of $k$ edge-disjoint paths from $\bot$ to $\top$ in $\mathbb{G}_T$. Let $M_T^1 = \cdots = M_T^k = 0$. We clearly have $M_T = 0 = \sum_{i=1}^k M_T^i$ and that $M_T^i = 0 \leq \log |p_T^i|$ for $1 \leq i \leq k$.

Suppose now that the inductive hypothesis holds for some $t > 1$. We proceed to show that it holds for $t - 1$. On trial $t - 1$ we receive example $(Z_{t-1}, y_{t-1})$. If $Z_{t-1} \in \{\bot, \top\}$ then since we are

"within" a PQ-game we have not made a mistake, so the inductive hypothesis holds trivially with $p_{t-1}^i := p_t^i$ and $M_{t-1}^i := M_t^i$ for $1 \le i \le k$. Suppose instead that $Z_{t-1} \notin \{\bot, \top\}$. Without loss of generality assume that the label $y_{t-1} = 1$. Define $\hat{p}_t^i$ to be equal to $p_t^i$ except that the "final" edge is removed i.e., $\hat{p}_t^i := p_t^i \cap \{(I, J) \in E(\mathbb{G}_t) : J \ne \top\}$. Since (by Proposition 5) $\mathbb{G}_t = \mathtt{merge}(\mathbb{G}_{t-1}, \uparrow\{Z_{t-1}\})$ observe that $\hat{p}_t^i \subset E(\mathbb{G}_{t-1})$. By construction observe that since $Z_{t-1} \notin V(\mathbb{G}_t)$, it is in no path $\hat{p}_t^i$ however there is at least one vertex $Z_{t-1}' \in V(\mathbb{G}_{t-1})$ and a $i'$ such that $\hat{p}_t^{i'} \cup \{(Z_{t-1}', Z_{t-1})\}$ is a directed path in $\mathbb{G}_{t-1}$. Define $r_{t-1}^\bot$ to be the longest directed path in $\mathbb{G}_{t-1}$ from $\bot$ to $Z_{t-1}$ and $r_{t-1}^\top$ to be the longest directed path in $\mathbb{G}_{t-1}$ from $Z_{t-1}$ to $\top$. Now define path $p_{t-1}^{i'} := \hat{p}_t^{i'} \cup \{(Z_{t-1}', Z_{t-1})\} \cup r_{t-1}^\top$. Finally select an arbitrary edge-disjoint extensions of the paths $\{\hat{p}_t^1, \ldots, \hat{p}_t^{i'-1}, \hat{p}_t^{i'+1}, \ldots, \hat{p}_t^k\}$ to paths $\{p_{t-1}^1, \ldots, p_{t-1}^{i'-1}, p_{t-1}^{i'+1}, \ldots, p_{t-1}^k\}$ so each is a path from $\bot$ to $\top$, in $\mathbb{G}_{t-1}$.

Now consider the sub-case where $\mathtt{longest\text{-}path}$ incurred a mistake in $\mathbb{G}_{t-1}$. Then we have $M_{t-1} = M_t + 1$ so choose $M_{t-1}^{i'} := 1 + M_t^{i'}$ and choose, for all $i \ne i'$, $M_{t-1}^i := M_t^i$. By the inductive hypothesis we hence have that $M_{t-1} = 1 + M_t = 1 + \sum_{i=1}^k M_t^i = \sum_{i=1}^t M_{t-1}^i$. We predicted $Z_{t-1}$ to be 0 so we hence have that $|r_{t-1}^\bot| \le |r_{t-1}^\top|$. Since $p_{t-1}^{i'} \setminus r_{t-1}^\top$ is a directed path in $\mathbb{G}_{t-1}$ from $\bot$ to $Z_{t-1}$ we have $|p_{t-1}^{i'} \setminus r_{t-1}^\top| \le |r_{t-1}^\bot|$ and we hence have that $|p_{t-1}^{i'} \setminus r_{t-1}^\top| \le |r_{t-1}^\top|$. Hence we have, by the inductive hypothesis, that $|p_{t-1}^{i'}| = |p_{t-1}^{i'} \setminus r_{t-1}^\top| + |r_{t-1}^\top| \ge 2|p_{t-1}^{i'} \setminus r_{t-1}^\top| = 2|p_t^{i'}| \ge 2 \times 2^{M_t^{i'}} = 2^{1+M_t^{i'}} = 2^{M_{t-1}^{i'}}$. Since, $|p_{t-1}^i| \ge |p_t^i|$ for all $i$, we have, by the inductive hypothesis, that, for all $i \ne i'$, $|p_{t-1}^i| \ge |p_t^i| \ge 2^{M_t^i} = 2^{M_{t-1}^i}$. And thus the sub-case with a mistake is shown.

Now consider the sub-case where we didn't make a mistake in $\mathbb{G}_{t-1}$. Then we have that $M_{t-1} = M_t$ so choose, for all $i$, $M_{t-1}^i := M_t^i$ and we have, by the inductive hypothesis, $M_{t-1} = M_t = \sum_{i=1}^k M_t^i = \sum_{i=1}^k M_{t-1}^i$. Since, for all $i$, $|p_{t-1}^i| \ge |p_t^i|$, we have, by the inductive hypothesis, that $|p_{t-1}^i| \ge |p_t^i| \ge 2^{M_t^i} = 2^{M_{t-1}^i}$. And thus the sub-case without a mistake is shown.

We conclude observing that the cumulative mistakes from trials 1 to $T$ of the PQ-game is $M_1 = \sum_{i=1}^k M_1^i \le \sum_{i=1}^k \log |p_1^i|$ $\qquad\square$

# D  Global mistake analysis (proofs of Theorems 4 and 10)

## D.1  Proof of Theorem 4

Suppose we have a uniformly-labeled subgraph $\mathcal{C}$.

A *covering sequence* is an example sequence (i.e. a sequence of pairs $(v, y)$ where $y$ is the label of vertex $v$) that contains every vertex in $\mathcal{G}$ and the label of any vertex in $\mathcal{C}$ is, without loss of generality, equal to 1.

When, given a covering sequence, $\mathcal{R}$, we say "mistakes made in $\mathcal{C}$ with $\mathcal{R}$" we mean "mistakes made in $\mathcal{C}$ when algorithm $\mathcal{A}$ is run on $\mathcal{R}$". We also say, for a vertex $w$, "$w$ is predicted 0 (resp. 1) with $\mathcal{R}$" when we mean "when algorithm $\mathcal{A}$ is run on $\mathcal{R}$ the label of $w$ is predicted, by $\mathcal{A}$, to be 0 (resp. 1).

**Lemma 15.** *Suppose we have a covering sequence $\mathcal{R} = \langle \mathcal{S}, (v, 1), \mathcal{T} \rangle$ for example sequences $\mathcal{S}$ and $\mathcal{T}$ where $v \notin \mathcal{C}$. Let $\mathcal{R}' := \langle \mathcal{S}, \mathcal{T}, (v, 0) \rangle$. Then the number of mistakes made in $\mathcal{C}$ with $\mathcal{R}'$ is at least the number of mistakes made in $\mathcal{C}$ with $\mathcal{R}$.*

*Proof.* All we need to show is that given some $w \in \mathcal{C}$ in which a mistake is made on $w$ (i.e. $w$ is predicted 0) with $\mathcal{R}$, then a mistake is made (i.e. $w$ is predicted 0) with $\mathcal{R}'$. This is clearly true if $(w, 1)$ is in $\mathcal{S}$ (the algorithms are identical on $\mathcal{S}$) so assume otherwise. We must then have that $(w, 1)$ is in $\mathcal{T}$. If $w$ is predicted 1 with $\mathcal{R}'$ then by monotonicity and permutation invariance, $w$ must be predicted 1 with $\mathcal{R}$ which is a contradiction. We must hence have that $w$ is predicted 0 with $\mathcal{R}'$. $\quad\square$

**Lemma 16.** *Suppose we have a covering sequence $\mathcal{R} = \langle \mathcal{S}, (v, 0), \mathcal{T} \rangle$ for example sequences $\mathcal{S}$ and $\mathcal{T}$ where $v \notin \mathcal{C}$. Let $\mathcal{R}' := \langle (v, 0), \mathcal{S}, \mathcal{T} \rangle$. Then the number of mistakes made in $\mathcal{C}$ with $\mathcal{R}'$ is at least the number of mistakes made in $\mathcal{C}$ with $\mathcal{R}$.*

*Proof.* All we need to show is that given some $w \in \mathcal{C}$ in which a mistake is made on $w$ (i.e. $w$ is predicted 0) with $\mathcal{R}$, then a mistake is made (i.e. $w$ is predicted 0) with $\mathcal{R}'$. This is clearly true if $(w, 1)$ is in $\mathcal{T}$ (the algorithms are identical on $\mathcal{T}$ by permutation invariance) so assume otherwise. We must then have that $(w, 1)$ is in $\mathcal{S}$. Since $w$ is predicted 0 with $\mathcal{R}$, by monotonicity and permutation invariance $w$ must be predicted 0 with $\mathcal{R}'$. $\qquad\square$

**Lemma 17.** *Given a covering sequence $\mathcal{R}$, there exists sequences $\mathcal{U}$ and $\mathcal{V}$ such that the elements of $\mathcal{U}$ are $\{(v, 0) : v \notin \mathcal{C}\}$ and the elements of $\mathcal{V}$ are $\{(v, 1) : v \in \mathcal{C}\}$ and at least as many mistakes are made in $\mathcal{C}$ with $\langle \mathcal{U}, \mathcal{V} \rangle$ as are made with $\mathcal{R}$.*

*Proof.* Repeatedly use Lemma 15 on $\mathcal{R}$ to form a covering sequence $\mathcal{R}'$ which makes at least as many mistakes in $\mathcal{C}$ as $\mathcal{R}$ and in which for every $v \notin \mathcal{C}$, the label of $v$ is 0. Next, repeatedly use Lemma 16 on $\mathcal{R}'$ to form the covering sequence $\langle \mathcal{U}, \mathcal{V} \rangle$ in the lemma. $\qquad\square$

Suppose then that $\mathcal{R}$ is our true label sequence. Then find sequences $\mathcal{U}$ and $\mathcal{V}$ as in Lemma 17. The number of mistakes made in $\mathcal{C}$ with sequence $\mathcal{R}$ is then no more than the number of mistakes made in $\mathcal{C}$ with sequence $\langle \mathcal{U}, \mathcal{V} \rangle$ which, by the Markov property, is bounded above by $\mathcal{B}_\mathcal{A}(\mathcal{C}; \mathcal{G})$. This completes the proof of Theorem 4. $\qquad\blacksquare$

### D.2 Proof Theorem 10

The proof of Theorem 10 separates into three cases: high-connectivity clusters, low-connectivity clusters, and a tree cluster. In each case we assume that we have received the label of each vertex in $\partial_e(\mathcal{C})$ and we then upper bound the number of mistakes in $\mathcal{C}$. Without loss of generality assume that each vertex of $\mathcal{C}$ is labelled 1 and each vertex of $\partial_e(\mathcal{C})$ is labelled 0.

**CASE 1 : If $\kappa(\mathcal{C}) > |\partial_e^E(\mathcal{C})|$ then $\mathcal{B}_\mathcal{A}(\mathcal{C}; \mathcal{G}) \in \mathcal{O}(1)$.**

*Proof.* Suppose we have made a single mistake in cluster $\mathcal{C}$. Then we have received the true label $y_v$ (equal to 1) for some vertex $v \in \mathcal{C}$. Let $\boldsymbol{u}$ be a consistent (with the observed labels) labelling of $\mathcal{C} \cup \partial_e(\mathcal{C})$ that minimises the cut. Note that we have $u_v = 1$ and for all $w \in \partial_e(\mathcal{C})$ we have $u_w = 0$. So if $u_z = 1$ for all $z \in \mathcal{C}$ then $\boldsymbol{u}$ has a cut of size $|\partial_e^E(\mathcal{C})|$. If there exists a vertex $z \in \mathcal{C}$ with $u_z = 0$ then, since there are at least $\kappa(\mathcal{C})$ edge disjoint paths between $z$ and $v$, we have that $\boldsymbol{u}$ has a cutsize of at least $\kappa(\mathcal{C})$. So since $\kappa(\mathcal{C}) > |\partial_e^E(\mathcal{C})|$ and $\boldsymbol{u}$ minimises the cut we must have that $u_z = 1$ for all $z \in \mathcal{C}$. Hence, since the the next prediction in $\mathcal{C}$ is consistent with a labelling of minimum-cut, it will predict the label as 1 so will not be a mistake. We can hence make at most one mistake in $\mathcal{C}$. $\qquad\square$

**CASE 2 : If $\kappa(\mathcal{C}) \le |\partial_e^E(\mathcal{C})|$ then $\mathcal{B}_\mathcal{A}(\mathcal{C}; \mathcal{G}) \in \mathcal{O}(|\partial_e^E(\mathcal{C})|(1 + |\partial_e^E(\mathcal{C})| - \kappa(\mathcal{C})) \log \mathcal{N}_{|\partial_e^E(\mathcal{C})|+1})$.**

*Proof.* We consider the sequence of PQ-games after we have made a single mistake. We first bound the cutsize of the first PQ-game. Since we have made a single mistake we have received the true label $y_v$ (equal to 1) for some vertex $v \in \mathcal{C}$. Let $\boldsymbol{u}$ be a consistent (with the observed labels) labelling of $\mathcal{C} \cup \partial_e(\mathcal{C})$ that minimises the cut. Note that we have $u_v = 1$ and for all $w \in \partial_e(\mathcal{C})$ we have $u_w = 0$. So if $u_z = 1$ for all $z \in \mathcal{C}$ then $\boldsymbol{u}$ has a cut of size $|\partial_e^E(\mathcal{C})|$. If there exists a vertex $z$ with $u_z = 0$ then, since there are at least $\kappa(\mathcal{C})$ edge disjoint paths between $z$ and $v$, we have that $\boldsymbol{u}$ has a cut of size at least $\kappa(\mathcal{C})$. Hence, since $\kappa(\mathcal{C}) \le |\partial_e^E(\mathcal{C})|$ we have that $\boldsymbol{u}$ has a cutsize of at least $\kappa(\mathcal{C})$. Hence, the cutsize of the first PQ-game is at least $\kappa(\mathcal{C})$.

Since the true cutsize is $|\partial_e^E(\mathcal{C})|$ we hence have that up to $|\partial_e^E(\mathcal{C})| - \kappa(\mathcal{C}) + 1$ PQ-games are played and the cutsize of each PQ-game is no greater than $|\partial_e^E(\mathcal{C})|$. We now bound the number of mistakes made in each PQ-game. To do this we first bound the number of super-vertices in any of the PQ-graphs. Suppose we have a set $X$ of vertices in $\mathcal{C}$ with connectivity greater than $|\partial_e^E(\mathcal{C})|$. Then suppose $\boldsymbol{u}$ is a consistent (with the observed labels, at any point in the algorithm) labelling of $\mathcal{C} \cup \partial_e(\mathcal{C})$ that minimizes the cut. We must have that $\boldsymbol{u}$ has a cutsize equal to the cutsize of a PQ-game and hence has a cutsize no greater than $|\partial_e^E(\mathcal{C})|$. Suppose, for contradiction, that there exist vertices $x, y \in X$ such that $u_x \ne u_y$. Then since there are at over $|\partial_e^E(\mathcal{C})|$ edge-disjoint paths from $x$ to $y$ we have that $\boldsymbol{u}$ has a cutsize greater than $|\partial_e^E(\mathcal{C})|$ which is a contradiction. We have just shown that for any consistent (with the observed labels) labelling, $\boldsymbol{u}$, of $\mathcal{C} \cup \partial_e(\mathcal{C})$ that minimizes

the cut we have that $u_x$ is identical for all $x \in X$. By definition of the PQ-graph this means that $X$ is a subset of some super-vertex of the PQ-graph. Hence, we have that any PQ-graph in the algorithm has at most $1+\mathcal{N}_{|\partial_e^E(\mathcal{C})|+1}(\mathcal{C})$ super-vertices (the "+1" corresponds to the super-vertex formed from $\partial_e(\mathcal{C})$).

Now we can apply Theorem 8 to sum the bounds of each PQ-game where we upper bound $k$ for each game by $|\partial_e^E(\mathcal{C})|$ and the path length by $\mathcal{N}_{|\partial_e^E(\mathcal{C})|+1}(\mathcal{C})$. Since there are at most $|\partial_e^E(\mathcal{C})| - \kappa(\mathcal{C}) + 1$ PQ-games this gives us a maximum of most $\mathcal{O}(|\partial_e^E(\mathcal{C})|(1 + |\partial_e^E(\mathcal{C})| - \kappa(\mathcal{C})) \log \mathcal{N}_{|\partial_e^E(\mathcal{C})|+1})$ mistakes inside the PQ-games. Finally, note that there are at most $|\partial_e^E(\mathcal{C})| - \kappa(\mathcal{C}) + 1$ mistakes between PQ-games. The result follows. □

**CASE 3 : If $\mathcal{C}$ is a tree then $\mathcal{B}_{\mathcal{A}}(\mathcal{C};\mathcal{G}) \in \mathcal{O}(|\partial_e^E(\mathcal{C})| \log D(\mathcal{C}))$.**

**Theorem 18.** *Given a tree structured subgraph, $\mathcal{C}$, of $\mathcal{G}$ we have $\mathcal{B}_{\mathcal{A}}(\mathcal{C};\mathcal{G}) \in \mathcal{O}(|\partial_e^E(\mathcal{C})| \log_2(D(\mathcal{C})))$*

**Proof of Theorem 18**

Suppose $k := \partial_e^E(\mathcal{C})$. For every vertex $v \in \mathcal{C}$ let $\eta(v)$ be the number of neighbours of $v$ that are not in $\mathcal{C}$. Then consider the tree $\mathcal{T}$ which is formed from $\mathcal{C}$ by adding, to each vertex $v$, $\eta(v)$ vertices. Label $\mathcal{T}$ as follows: if $v \in \mathcal{C}$ label $v$ as 1 and if $v \notin \mathcal{C}$ label $v$ as 0. Then $\mathcal{B}_{\mathcal{A}}(\mathcal{C};\mathcal{G})$ is upper-bounded by the maximum number of mistakes made in $\mathcal{T}$ with any permutation of the vertices. Note that the cutsize of the labelling of $\mathcal{T}$ is equal to $k$ so by Appendix F, the number of mistakes made in $\mathcal{T}$ is upper-bounded by $\mathcal{O}(\text{LB}(\mathcal{T}, k))$ which is upper bounded by $\mathcal{O}(k \log(D(\mathcal{T})))$. The result follows since $D(\mathcal{T}) \leq D(\mathcal{C}) + 2$. □

# E Regularity properties of `longest-path` and `0-Ising` (proof of Theorem 9)

The proof of all properties except for the label-monotonicity of `longest-path` are straightforward.

## E.1 Proof that `longest-path` is label-monotone

In this proof we use the more explicit notation $(v \to w)$ for a directed edge from $v$ to $w$.

Let $\mathcal{S}$ be an example sequence. Let $z$ be a vertex of $\mathcal{G}$ that is not contained in $\mathcal{S}$. Define $\mathcal{S}' := \langle \mathcal{S}, (z, 0) \rangle$ (note that, in what follows, we don't lose generality by assuming that $z$ is labelled 0 by the symmetry over switching the labels 0 and 1 on all vertices). Let $(\mathcal{H}, s, t)$ (resp. $(\mathcal{H}', s', t')$) be the result of step 1 of the PQ-graph construction algorithm (see Figure 1) when run on sequence $\mathcal{S}$ (resp. $\mathcal{S}'$). Note that $\mathcal{H}'$ is identical to $\mathcal{H}$ except that the vertices $s$ and $z$ are merged into a single vertex $s'$. Let $\mathbb{G}$ (resp. $\mathbb{G}'$) be the graph formed at step 4 of the PQ-graph construction algorithm (see Figure 1) when run on sequence $\mathcal{S}$ (resp. $\mathcal{S}'$). Let $\perp$ and $\top$ (resp. $\perp'$ and $\top'$) be the source and target super-vertices of $\mathbb{G}$ (resp. $\mathbb{G}'$) respectively. Given $v \in V(\mathcal{H})$ (resp. $v \in V(\mathcal{H}')$) define $\Psi(v)$ (resp. $\Psi'(v)$) to be the super-vertex in $\mathbb{G}$ (resp. $\mathbb{G}'$) that contains $v$.

To prove label monotonicity we need to show that given $h \in V(\mathcal{H}') \setminus \{s', t'\}$, if the label of $h$ is predicted 0 in $\mathbb{G}$ then it is predicted 0 in $\mathbb{G}'$. Let $\phi$ and $\psi$ be the longest directed paths in $\mathbb{G}$ from $\perp$ to $\Psi(h)$ and from $\Psi(h)$ to $\top$ respectively. Let $\phi'$ and $\psi'$ be the longest directed paths in $\mathbb{G}'$ from $\perp'$ to $\Psi'(h)$ and from $\Psi'(h)$ to $\top'$ respectively. If $\Psi'(h) = \perp'$ then we are done since then the label of $h$ is predicted 0 in $\mathbb{G}'$, so assume otherwise. Since the label of $h$ is 0 in $\mathbb{G}$ we also have that $h \notin \top$.

We need the following proposition about the graphs $\mathbb{G}$ and $\mathbb{G}'$. We will prove the proposition later in the section.

**Proposition 19.** *We have the following results:*

*if* $X, Y \in V(\mathbb{G}'), X \subseteq \top$ *and* $(X \to Y) \in E(\mathbb{G}')$ *then* $Y \subseteq \top$ **[edge creation]** $\qquad$ (a)

*if* $v \in V(\mathcal{H}) \setminus (\top \cup \{s,z\})$ *then* $\Psi'(v) = \bot'$ *or* $\Psi'(v) = \Psi(v)$ **[vertex collapse/conservation]**
$\qquad$ (b)

*if* $v, w \in V(\mathcal{H}) \setminus (\top \cup \{s,z\})$ *and* $\Psi'(v) \neq \bot'$ *then* $\qquad\qquad$ **[edge conservation]**
$$(\Psi'(v) \to \Psi'(w)) \in E(\mathbb{G}') \Leftrightarrow (\Psi(v) \to \Psi(w)) \in E(\mathbb{G}) \qquad \text{(c)}$$

**Lemma 20.** $|\phi'| \leq |\phi|$

*Proof.* Since, $\Psi'(h) \neq \bot'$ write $\phi'$ as $(\bot', X_1, X_2, ..., X_m = \Psi'(h))$. If, for some $i$, we have $X_i \subseteq \top$ then by Proposition 19 Item (a) (by induction through $X_i, X_{i+1}, ..., X_m = \Psi'(h)$ using inductive hypothesis $X_j \subseteq \top$) we would have $\Psi'(h) \subseteq \top$ which would imply that $h \in \top$ which is a contradiction. Also, if for some $i$ we have $s' \in X_i$, we would have $X_i = \bot'$ which is a contradiction. Hence, for every $i$, we have some vertex $x_i \in \mathcal{H} \setminus (\top \cup \{s,z\})$ such that $X_i = \Psi'(x_i)$. We hence have a path $(\Psi'(x_1), \Psi'(x_2), ..., \Psi'(x_m))$ in $\mathbb{G}'$ where, for all $i$, $x_i \in \mathcal{H} \setminus (\top \cup \{s,z\})$ and $\Psi'(x_i) \neq \bot'$. By Proposition 19 Item (b) we have that for each $i$, $\Psi'(x_i) = \Psi(x_i)$ (and since $h \in \mathcal{H} \setminus (\top \cup \{s,z\})$, we have $\Psi'(x_m) = \Psi'(h) = \Psi(h)$) so by Proposition 19 Item (c) $(\Psi(x_1), \Psi(x_2), ..., \Psi(x_m) = \Psi(h))$ is a directed path in $\mathbb{G}$. Since $\Psi(x_1) = \Psi'(x_1)$ and (since $\Psi'(x_1)$ is a subset of $\mathcal{H}'$) $s \notin \Psi'(x_1)$ we must have $\Psi(x_1) = \Psi'(x_1) \neq \Psi(s) = \bot$. We can hence continue (in $\mathbb{G}$) the path $(\Psi(x_1), \Psi(x_2), ..., \Psi(x_m) = \Psi(h))$ back to $\bot$ giving us, for some $m' \geq 0$ a directed path $(\bot, Y_1, Y_2, ...Y_{m'}, \Psi(x_1), \Psi(x_2), ..., \Psi(x_m) = \Psi(h))$ in $\mathbb{G}$. We hence have constructed a directed path in $\mathbb{G}$, from $\bot$ to $\Psi(h)$ that is at least as long as $\phi'$ which proves the result. $\qquad \square$

**Lemma 21.** $|\psi| \leq |\psi'|$

*Proof.* Since $h \notin \top$ and hence $\Psi(h) \neq \top$ write $\psi$ as $(\Psi(h) = X_1, X_2, ..., X_m, \top)$. Let $x_1 := h$ and for $i > 2$ let $x_i$ be an arbitrary member of $X_i$. Since $h \notin \top$ and $h \neq s, z$ we have $x_1 \in \mathcal{H} \setminus (\top \cup \{s,z\})$ For $i > 2$ we know (since there is no edge in $\mathbb{G}$ that goes into $\bot$) that $X_i \neq \bot$ and hence $s \notin X_i$. Since $X_i \neq \top$ we then have that $x_i \in \mathcal{H} \setminus (\top \cup s)$. Hence, for all $i$ we have $x_i \in \mathcal{H} \setminus (\top \cup s)$. Suppose, for contradiction, that, for some $i$, $x_i = z$. Then $z \in X_i \neq \top$ so $z \notin \top$. Hence, by Proposition 5 and since $\Psi(h)$ is downstream of $X_i$ we have that $h \in \bot'$ which contradicts the assumptions of $h$. Hence we have that, for all $i$, $x_i \neq z$ and hence $x_i \in \mathcal{H} \setminus (\top \cup \{s,z\})$.

Note that $(\Psi(x_1), \Psi(x_2), ..., \Psi(x_m))$ is a directed path in $\mathbb{G}$ (as it is a subpath of $\psi$). We now prove the following by induction on $i$:

1. $\Psi'(x_i) \neq \bot'$

2. $\Psi'(x_i) = \Psi(x_i)$

3. $(\Psi'(x_i) \to \Psi'(x_{i+1}))$ is an edge in $\mathbb{G}'$ (for $i \neq m$)

Note that items 2 and 3 can be proved from Item 1 as follows: Since $x_i \in \mathcal{H} \setminus \{\top \cup \{s,z\}\}$ and $\Psi'(x_i) \neq \bot'$ then by Proposition 19 Item (b) we have $\Psi'(x_i) = \Psi(x_i)$. Since $x_i, x_{i+1} \in \mathcal{H} \setminus \{\top \cup \{s,z\}\}$ and $\Psi'(x_i) \neq \bot'$ then by Proposition 19 Item (c) we have that $(\Psi'(x_i) \to \Psi'(x_{i+1}))$ is an edge in $\mathbb{G}'$.

We now prove Item 1 from the inductive hypothesis: For $i = 1$ we have (since $h \notin \bot'$) that $\Psi'(x_1) = \Psi'(h) \neq \bot'$. For $i > 1$ we have, from the inductive hypothesis, that $(\Psi'(x_{i-1}) \to \Psi'(x_i))$ is an edge in $\mathbb{G}'$. But no edge in $\mathbb{G}'$ goes into $\bot'$, so $\Psi'(x_i) \neq \bot'$. This completes the inductive proof of the above items.

We hence have that $(\Psi'(x_1), \Psi'(x_2), ..., \Psi'(x_m))$ is a directed path in $\mathbb{G}'$. Note that since $x_m \notin \top$ we have $t' = t \notin \Psi(x_m)$ so (by Item 2 above) we have $t' \notin \Psi(x_m) = \Psi'(x_m)$ and hence $\Psi'(x_m) \neq \top'$. So we can extend (in $\mathbb{G}'$) the path $(\Psi'(x_1), \Psi'(x_2), ..., \Psi'(x_m))$ to a path $(\Psi'(h) = \Psi'(x_1), \Psi'(x_2), ..., \Psi'(x_m), Y_1, Y_2, Y_{m'}, \top')$ for some $m' \geq 0$. We have now constructed a path in $\mathbb{G}'$ from $\Psi'(h)$ to $\top'$ that is at least as long as $|\psi|$ which proves the result. $\qquad \square$

Since the label of $h$ was predicted as 0 in $\mathcal{H}$ we have that $|\phi| \leq |\psi|$. Hence, by Lemmas 20 and 21 we have that $|\phi'| \leq |\phi| \leq |\psi| \leq |\psi'|$. So $|\phi'| \leq |\psi'|$ implying that the label of $h$ is still predicted as 0 in $\mathcal{H}'$. ∎

### E.1.1 Proof of Proposition 19

By Proposition 5, Proposition 19 clearly holds if $z \notin \top$ so assume otherwise. Let $K$ (resp. $K'$) be the cut-size of a label-consistent minimum-cut of $\mathcal{H}$ (resp. $\mathcal{H}'$). Let $B = \{(x \to y) : (x, y) \in \mathcal{H}, x \notin \top, y \in \top\}$. Given a flow in a graph, we define the *size* of the flow to be the number of edge disjoint paths in it.

We now construct the flow $\mathcal{F}$ (in step 2 of the PQ-graph construction algorithm for $\mathbb{G}$) from $s$ to $t$ in $\mathcal{H}$ of size $K$, and the flow $\mathcal{F}'$ (in step 2 of the PQ-graph construction algorithm for $\mathbb{G}'$) from $s'$ to $t'$ in $\mathcal{H}'$ of size $K'$ as follows (note that we will refer to the objects in this algorithm later):

**Algorithm 22.**

1. *Convert $\mathcal{H}$ to a graph $\mathcal{H}''$ by adding a set $A$ of $K'$ edges between $s$ and $z$.*

2. *By running the Ford-Fulkerson algorithm on $\mathcal{H}$ construct a flow, $\mathcal{F}$, of size $K$ in $\mathcal{H}''$ from $s$ to $t$ such that none of the edges in $A$ are contained in the flow. Note that this is the first $K$ steps in an instance of the Ford-Fulkerson algorithm on $\mathcal{H}''$. Note also that $\mathcal{F}$ is a flow of size $K$ in $\mathcal{H}$.*

3. *Continue (from stage 2) the Ford-Fulkerson algorithm on $\mathcal{H}''$ to get a flow, $\mathcal{F}''$, of size $K'$ from $s$ to $t$ in $\mathcal{H}''$.*

4. *Set $\mathcal{F}'''$ equal to $\mathcal{F}''$. Repeat the following until there is no directed path in $\mathcal{F}'''$ from $s$ to $z$ that does not contain an edge in $A$:*

   (a) *Choose a directed path in $\mathcal{F}'''$ from $s$ to $z$ that does not contain an edge in $A$. Remove this path from $\mathcal{F}'''$ and add to $\mathcal{F}'''$ an edge in $A$ (directed from $s$ to $z$). Note that $\mathcal{F}'''$ is still a valid flow of size $K'$.*

5. *Merge the vertices $s$ and $z$ to get, from $\mathcal{F}'''$, a flow, $\mathcal{F}'$, of size $K'$ from $s'$ to $t'$ in $\mathcal{H}'$.*

We now let $\mathcal{I}$ and $\mathcal{I}'$ be the graphs formed in step 3 of the PQ-graph construction algorithm for $\mathbb{G}$ (given the maximum flow $\mathcal{F}$) and $\mathbb{G}'$ (given the maximum flow $\mathcal{F}'$) respectively.

**Lemma 23.** *For all $x, y \in \mathcal{H}$, if $(x \to y) \in B$ then we have $(x \to y) \in \mathcal{F}$.*

*Proof.* Assume we have some $x, y \in \mathcal{H}$, with $(x \to y) \in B$. Suppose, for contradiction, that both $(x \to y)$ and $(y \to x)$ are not in $\mathcal{F}$. Then we have that $(x \to y)$ and $(y \to x)$ are both in $\mathcal{I}$ implying that $\Psi(x) = \Psi(y)$. Then since $y \in \top$ we have $\Psi(x) = \Psi(y) = \top$, which contradicts the fact that $x \notin \top$. We hence have that either $(x \to y)$ or $(y \to x)$ are in $\mathcal{F}$. Suppose, now, for contradiction, that $(y \to x) \in \mathcal{F}$. Then it is a result of the Ford-Fulkerson algorithm that there exists a directed path $p$, in $\mathcal{F}$, from $y$ to $t$ that goes through $x$. Let $p = (y = v_1, v_2, ...v_m = t)$. Since, for all $i$ we have $(v_i \to v_{i+1}) \in \mathcal{F}$, it is the case that $(v_{i+1} \to v_i) \notin \mathcal{F}$ so $(v_i \to v_{i+1}) \in \mathcal{I}$. Hence $p$ is a path in $\mathcal{I}$ from $t$ to $y$ that goes through $x$. Since $y \in \top$ and hence $\Psi(y) = \top = \Psi(t)$ we have a directed path in $\mathcal{I}$ from $y$ to $t$. Putting these together we hence have a directed cycle in $\mathcal{I}$ that contains $t$ and $x$. So $t$ and $x$ are in the same strongly connected component of $\mathcal{I}$ and hence $\Psi(x) = \Psi(t) = \top$. This contradicts the fact that $x \notin \top$. We hence have that $(y \to x) \notin \mathcal{F}$ which, by the above, implies that $(x \to y) \in \mathcal{F}$. □

**Definition 24.** *For $0 \leq a \leq (K' - K)$, let $\mathcal{F}_a$ be the flow during step $a$ of stage 3 of algorithm 22 (i.e. $\mathcal{F}_0 = \mathcal{F}$, $\mathcal{F}_{K'-K} = \mathcal{F}''$, and for all $a$, the size of $\mathcal{F}_{a+1}$ is one more than the size of $\mathcal{F}_a$). For every $0 \leq a \leq (K' - K)$ define $C_a := \mathcal{F}_a \setminus (\{(x \to y) : x, y \in \top\} \cup A)$.*

**Lemma 25.** *For all $0 \leq a \leq (K' - K)$ we have $C_a = C_0$.*

*Proof.* We prove by induction on $a$. The inductive hypothesis clearly holds for $a = 0$.

Now suppose the inductive hypothesis holds for some $a$. We now show that it also holds for $a + 1$:

First note that by Lemma 23 every directed edge in $B$ is contained in $C_0$. Now, let $p$ be the path (from $s$ to $t$) that is found in the Ford-Fulkerson algorithm when the flow goes from $\mathcal{F}_a$ to $\mathcal{F}_{a+1}$. Then let $(s, z = x_0, x_1, x_2, \ldots x_m = t) := p$ (where the edge $(s, z)$ is in $A$)

Suppose, for contradiction, that for some $i \le m$, $x_i \notin \top$. Then let $j := \min\{i : x_i \notin \top\}$. Let $k := \min\{i > j : x_i \in \top\}$ which is defined since $x_m = t \in \top$. We have $(x_{k-1} \to x_k) \in B$ so $(x_{k-1} \to x_k) \in C_0$ and hence by the inductive hypothesis $(x_{k-1} \to x_k) \in C_a$ so $(x_{k-1} \to x_k) \in \mathcal{F}_a$ which contradicts the fact that $p$ is the path found by the Ford-Fulkerson algorithm.

Hence, all the edges in $p$ are in $\{(x \to y) : x, y \in \top\} \cup A$ and hence, by considering the Ford-Fulkerson algorithm, we have $C_{a+1} = C_a$. Hence, by the inductive hypothesis we have $C_{a+1} = C_0$. $\qquad\square$

**Definition 26.** *Let $J := \mathcal{F} \setminus \{(x \to y) : x, y \in \top\}$ and $J' := \mathcal{F}' \setminus \{(x \to y) : x, y \in \top\}$.*

**Lemma 27.** *We have the following results:*

1. *Given $(x \to y) \in J'$, either $(x \to y) \in J$ or $x = s'$.*

2. *Given $(x \to y) \in J$, either $(x \to y) \in J'$ or $x, y \in \bot'$ or $x \in \{s, z\}$ or $y \in \{s, z\}$.*

*Proof.* Let $J'' := \mathcal{F}'' \setminus (\{(x \to y) : x, y \in \top\} \cup A)$. Note that $J = C_0$ and $J'' = C_{K'-K}$ so by Lemma 25 we have that $J'' = J$.

Suppose we have $(x \to y) \in J'$ with $x \ne s'$. Then we automatically have that $(x \to y) \in \mathcal{F}'''$ at the start of stage 5 of algorithm 22. On each step in stage 4 of algorithm 22 the only edges added to $\mathcal{F}'''$ are those in $A$ so since (because $x, y \in \mathcal{H}'$ so $x, y \ne s$) $(x, y) \notin A$ we have that $(x \to y) \in \mathcal{F}''$. Since $(x \to y) \notin \{(v \to w) : v, w \in \top\} \cup A$ we hence have $(x \to y) \in J''$ which implies, by the above, that $(x \to y) \in J$. This proves Item 1 of the lemma.

Suppose we have some $(x \to y) \in J$ with $(x \to y) \notin J'$, $x, y \notin \{s, z\}$. Since $(x \to y) \in J$ we have (since, by the above, $J = J''$) that $(x \to y) \in J''$. Since $(x \to y) \in J''$ we have $(x \to y) \notin \{(v \to w) : v, w \in \top\}$ and hence, since $(x \to y) \notin J'$, $(x \to y) \notin \mathcal{F}'$. Since $(x \to y) \in J''$ we must have $(x \to y) \in \mathcal{F}''$. So $(x \to y) \in \mathcal{F}''$ and $(x \to y) \notin \mathcal{F}'$ and hence (since $x, y \notin \{s, z\}$) it must be the case that $(x \to y)$ was removed (from $\mathcal{F}'''$) during stage 4 of algorithm 22. Let $(s = v_1, v_2, \ldots, v_m = z)$ be the directed path in $\mathcal{F}''$ that contains $(x \to y)$ and was removed during stage 4 of algorithm 22. Since this path is removed and $s$ and $z$ are merged into $s'$ in forming $\mathcal{F}'$ we have that no edge in the cycle (in $\mathcal{H}'$) $(s', v_2, v_3, \ldots, v_{m-1}, s')$ is in $\mathcal{F}'$. Hence we have that $(s', v_{m-1}, v_{m-1}, \ldots, v_2, s')$ is a directed cycle in $\mathcal{I}'$ so all vertices in this cycle belong to the same strongly connected component of $\mathcal{I}'$. This implies that for all $i$ we have $\Psi'(v_i) = \Psi'(s') = \bot'$. Since $x, y \notin \{s, z\}$ we have that $x = v_i$ and $y = v_{i+1}$ for some $1 < i < m - 1$. Hence we have that $x, y \in \bot'$ which completes the proof of item 2 of the lemma. $\qquad\square$

**Lemma 28.** *Given some $v \in \top$ with $v \ne z$ we either have $\Psi'(v) = \bot'$ or $\Psi'(v) \subseteq \top$.*

*Proof.* Suppose we assume the converse: that there exists some $v \in \top \setminus \{z\}$ with $\Psi'(v) \ne \bot'$ and $\Psi'(v) \not\subseteq \top$. Then choose some $x \in \mathcal{H}' \setminus \top$ such that $x \in \Psi'(v)$. Since $x$ and $v$ are in the same strongly connected component in $\mathcal{I}'$ there exists (in $\mathcal{I}'$) a directed path $p := \{v = x_0, x_1, x_2, \ldots, x_m = x\}$ such that each $x_i$ is in $\Psi'(v)$. Let $i = \min\{j : x_j \notin \top\}$ which exists since $x_m \notin \top$. Note that since $x_0 = v \in \top$ we have $i > 0$ so $x_{i-1}$ exists. Since $\Psi'(v) \ne \bot'$ we know $x_i \ne s'$ (else $s' \in \Psi'(v)$ and hence $\Psi'(v)$ would equal $\bot'$). We hence have that $(x_i \to x_{i-1}) \in B$ so we know, from Lemma 23 that $(x_i \to x_{i-1})$ is in $J$.

Since $(x_{i-1} \to x_i)$ is a directed edge in $p$, and hence in $\mathcal{I}'$, we know that $(x_i \to x_{i-1}) \notin \mathcal{F}'$ so we have that $(x_i \to x_{i-1}) \notin J'$ which implies by Lemma 27 Item 2 (since, by the above, $(x_i \to x_{i-1})$ is in $J$ and (since $x_i, x_{i-1} \in \mathcal{H}'$) $x_i, x_{i-1} \notin \{s, z\}$) that $x_i \in \bot'$. Since $x_i \in \Psi'(v)$ this implies that $\Psi'(v) = \bot'$ which is a contradiction. $\qquad\square$

**Lemma 29.** *Given some $X, Y \in \mathbb{G}'$ with $X \subseteq \top$, if there is an edge in $\mathbb{G}'$ from $X$ to $Y$, then we have $Y \subseteq \top$.*

*Proof.* Suppose the converse: that there exists some $X, Y \in \mathbb{G}'$ with $X \subseteq \top$, $Y \not\subseteq \top$ and an edge in $\mathbb{G}'$ from $X$ to $Y$.

Note first that since $X \subseteq \top$ we have $s' \notin X$ and hence $X \neq \bot'$. Since there is an edge in $\mathbb{G}'$ from $X$ to $Y$ and no edge goes into $\bot'$ we have $Y \neq \bot'$.

Since there is an edge in $\mathbb{G}'$ from $X$ to $Y$ choose $x \in X$ and $y \in Y$ such that there is an edge in $\mathcal{F}'$ from $x$ to $y$

Since $Y \not\subseteq \top$ and $Y \neq \bot'$ and (since $y \in \mathcal{H}'$) $y \neq z$ we must have, by Lemma 28, that $y \notin \top$. Hence we have that $(x \to y) \in J'$. Since $Y \neq \bot'$ we also have that $y \neq s'$. We hence have that $(y \to x) \in B$ so, by Lemma 23, we have that $(y \to x) \in J$. By Lemma 27 Item 1 we hence have a contradiction (since $(y \to x) \in J$ implies $(x \to y) \notin J$ and we have $(x \to y) \in J'$ and $x \neq s'$). $\qquad\square$

**Lemma 30.** *Given a vertex $v$ such that $(v \to s) \in \mathcal{I}$ and $v \notin \top$ we have that $(v \to s') \in \mathcal{I}'$.*

*Proof.* Suppose the converse: that there exists a vertex $v$ such that $(v \to s) \in \mathcal{I}$, $v \notin \top$ and $(v \to s') \notin \mathcal{I}'$. Since $(v \to s') \notin \mathcal{I}'$ we have $(s' \to v) \in \mathcal{F}'$. Hence, by considering Stage 5 of Algorithm 22 we must have that either $(s \to v) \in \mathcal{F}'''$ or $(z \to v) \in \mathcal{F}'''$. Since $v \notin \top$ we have $v \neq z$ so (since $v \neq s$ as there is an edge in $\mathcal{I}$ from $s$ to $v$) we have $(s,v), (z,v) \notin A$. Hence, since during Stage 4 of Algorithm 22 the only edges added to the flow are those in $A$, we must have that either $(s \to v) \in \mathcal{F}''$ or $(z \to v) \in \mathcal{F}''$.

Assume, for contradiction, that $(z \to v) \in \mathcal{F}''$. Then $(z,v) \in E(\mathcal{H}'')$ so since, by the above, $(z,v) \notin A$ (and, since $z, v \in \mathcal{H}$ we have $z, v \neq s'$) we have that $(z,v) \in E(\mathcal{H})$. Since $z \in \top$ and $v \notin \top$ we have that $(v \to z) \in B$ and hence, by Lemma 23, we have that $(v \to z) \in C_0$ so, by Lemma 25, $(v \to z) \in C_{K'-K}$ which implies that $(v \to z) \in \mathcal{F}''$ and hence that $(z \to v) \notin \mathcal{F}''$ which is a contradiction.

We hence have that $(s \to v) \in \mathcal{F}''$ which implies, since $v \notin \top$, that $(s \to v) \in C_{K'-K}$ so by Lemma 25 we have $(s \to v) \in C_0$ which implies that $(s \to v) \in \mathcal{F}$. This implies that $(v \to s) \notin \mathcal{I}$ which is a contradiction. This completes the proof. $\qquad\square$

**Lemma 31.** *Given some $v \in \mathcal{H} \setminus (\top \cup \{s\})$ with $\Psi'(v) \neq \bot'$ or $\Psi(v) \neq \bot$, we have $\Psi(v) \subseteq \Psi'(v)$.*

*Proof.* We shall prove that $\Psi(v)$ is strongly connected in $\mathcal{I}'$ which directly implies the result.

We first show that $s$ and $z$ are not contained in $\Psi(v)$. Since $v \notin \top$ we have $\Psi(v) \neq \top = \Psi(z)$ which implies that $z \notin \Psi(v)$. Suppose now, for contradiction, that $s \in \Psi(v)$. Since $v \neq s$ we then have a directed paths $(s, x_1, x_2, ..., x_m := v)$ and $(v = y_1, y_2, ..., y_{m'}, s)$ in $\mathcal{I}$ such that, for all $i$, $x_i, y_i \in \Psi(v)$. Note that since $z \notin \Psi(v)$ none of the $x_i$ or $y_i$ are equal to $z$. Since $(s, x_1)$ is an edge in $\mathcal{H}$ we must then have that $(s', x_1)$ is an edge in $\mathcal{H}'$ and since no edge of $\mathcal{F}'$ goes into $s'$ we must have that $(x_1 \to s') \notin \mathcal{F}'$ implying that $(s' \to x_1) \in \mathcal{I}'$. Since $(y_{m'} \to s) \in \mathcal{I}$ and (since $\Psi(y_{m'}) = \Psi(v) \neq \top$) $y_{m'} \notin \top$ we have, by Lemma 30 that $(y_{m'} \to s') \in \mathcal{I}'$. Since, for all $i$, $(x_i \to x_{i+1}) \in \mathcal{I}$ we have that $(x_{i+1} \to x_i) \notin \mathcal{F}$ hence $(x_{i+1} \to x_i) \notin J$. If it was true that $(x_{i+1} \to x_i) \in \mathcal{F}'$ then since $x_i \notin \top$ (as $v \notin \top$ implies that $\Psi(x_i) = \Psi(v) \neq \top$) we would have that $(x_{i+1} \to x_i) \in J'$ so since $x_{i+1}, x_i \neq s'$ (as both are in $\mathcal{H}$) we would have, by Lemma 27 Item 1, that $(x_{i+1} \to x_i) \in J$ which is a contradiction. Hence we have that $(x_{i+1} \to x_i) \notin \mathcal{F}'$ so $(x_i \to x_{i+1}) \in \mathcal{I}'$. Similarly we have $(y_i \to y_{i+1}) \in \mathcal{I}'$ for all $i$. We hence have that $(s', x_1, x_2, ...x_m = v)$ and $(v = y_1, y_2, ..., y_{m'}, s')$ are directed paths in $\mathcal{I}'$ so we have that $v$ and $s'$ are in the same strongly connected component of $\mathcal{I}'$. So $\Psi'(v) = \Psi(s') = \bot'$ and hence $v \in \bot'$. Since $s \in \Psi(v)$ we also have that $\Psi(v) = \bot$ which is a contradiction.

We hence have that $s, z \notin \Psi(v)$ so $\Psi(v) \subseteq V(\mathcal{H}')$. Suppose that we have vertices $x, y \in \Psi(v)$. We now show that there exists a directed path in $\mathcal{I}'$ from $x$ to $y$ which proves that $\Psi(v)$ is strongly connected in $\mathcal{I}'$:

Since $\Psi(v)$ is strongly connected in $\mathcal{I}$ there exists a directed path $p$ from $x$ to $y$ in $\mathcal{I}$ such that every vertex in $p$ is in $\Psi(v)$. Since $\Psi(v) \neq \top$, $p$ is a path in $V(\mathcal{H}) \setminus \top$. Hence, if some directed edge $(x' \to y')$ is in $p$ and not in $\mathcal{I}'$ then (since, by definition of $\mathcal{I}'$, $(y' \to x') \in \mathcal{F}'$) we have (since $x' \in \Psi(v) \neq \top$ and hence $x' \notin \top$) that $(y' \to x') \in J'$ which implies, by Lemma 27 Item 1 (since $y' \in \mathcal{H}$ so $y' \neq s'$) that $(y' \to x') \in J$, and hence $(y' \to x') \in \mathcal{F}$, which implies that $(x' \to y')$ is not in $\mathcal{I}$ which is a contradiction. Hence, $p$ is a directed path in $\mathcal{I}'$. This completes the proof that $\Psi(v)$ is strongly connected in $\mathcal{I}'$. The result follows. $\qquad\square$

**Lemma 32.** *Given some $v \in \mathcal{H} \setminus (\top \cup \{s\})$ we either have $\Psi'(v) = \bot'$ or $\Psi'(v) = \Psi(v)$.*

*Proof.* Suppose the converse: that there exists some $v \in \mathcal{H} \setminus (\top \cup \{s\})$ with $\Psi'(v) \neq \bot'$ and $\Psi'(v) \neq \Psi(v)$. Since $\Psi'(v) \neq \bot'$, we have, by Lemma 31, that $\Psi(v) \subseteq \Psi'(v)$. Since $\Psi(v) \neq \Psi'(v)$ we hence can choose some $x \in \Psi'(v) \setminus \Psi(v)$. Since $x \in \Psi'(v)$ we have a directed path (in $\mathcal{I}'$), $p$ (resp. $q$) in $\Psi'(v)$ from $v$ to $x$ (resp. $x$ to $v$).

Suppose, for contradiction, that there exists some $v' \in \Psi'(v)$ with $v' \in \top$. By the above we have $\Psi'(v') = \Psi'(v) \neq \bot'$. Note also that since $v' \in \mathcal{H}'$ we have $v' \neq z$. Hence, by Lemma 28 we must have that $\Psi'(v') \subset \top$ which implies (since $\Psi'(v) = \Psi'(v')$) that $\Psi'(v) \subset \top$ which contradicts the fact that $v \in \Psi'(v)$. Hence we have that no element of $\top$ is contained in $\Psi'(v)$.

Since $\Psi'(v) \neq \bot'$ no element of $\Psi'(v)$ is equal to $s'$. We hence have that the path $p$ contains only vertices in $\mathcal{H}' \setminus (\top \cup \{s'\})$. Hence, if some directed edge $(x' \to y')$ is in $p$ and not in $\mathcal{I}$ then (since, by definition of $\mathcal{I}$, $(y' \to x') \in \mathcal{F}$) we have $(y' \to x') \in J$ which implies, by Lemma 27 Item 2 that $(y' \to x') \in J'$ (because else, by Lemma 27 Item 2, $y', x' \in \bot'$ (since $y', x' \in \mathcal{H}'$ and hence $y', x' \notin \{s, z\}$) which is a contradiction since $y', x' \in \Psi'(v) \neq \bot'$) and hence $(y' \to x') \in \mathcal{F}'$, which implies that $(x' \to y')$ is not in $\mathcal{I}'$ which is a contradiction. Hence, $p$ is a directed path in $\mathcal{I}$. Similarly $q$ is a directed path in $\mathcal{I}$. This implies that $v$ and $x$ are in the same strongly connected component of $\mathcal{I}$. Hence $\Psi(v) = \Psi(x)$ so $x \in \Psi(v)$ which is a contradiction. $\square$

**Lemma 33.** *Given some $v, w \in \mathcal{H} \setminus (\top \cup \{s\})$ in which $\Psi'(v) \neq \bot'$ then the existence of an edge in $\mathbb{G}'$ from $\Psi'(v)$ to $\Psi'(w)$ implies the existence of an edge in $\mathbb{G}$ from $\Psi(v)$ to $\Psi(w)$.*

*Proof.* Note first that since there is an edge in $\mathbb{G}'$ going into $\Psi'(w)$ we must have $\Psi'(w) \neq \bot'$. By Lemma 32 we then have that $\Psi'(v) = \Psi(v)$ and $\Psi'(w) = \Psi(w)$. Since there is an edge in $\mathbb{G}'$ from $\Psi'(v)$ to $\Psi'(w)$ there exist vertices $x \in \Psi'(v)$ and $y \in \Psi'(w)$ such that $(x \to y) \in \mathcal{F}'$. Since $\Psi'(v) \neq \bot'$ we have $x \neq s'$. Since $\Psi'(x) = \Psi'(v) \neq \bot'$ and (as $v \in \Psi'(v) = \Psi'(x)$ and $v \notin \top$) $\Psi'(x) \not\subseteq \top$ we have, by Lemma 28, that $x \notin \top$. We hence have that $(x \to y) \in J'$ and that $x \neq s'$ so, by Lemma 27 Item 1, we have $(x \to y) \in J$ and hence there is an edge from $x$ to $y$ in $\mathcal{F}$. Since $\Psi'(v) = \Psi(v)$ and $\Psi'(w) = \Psi(w)$, we hence obtain the result (since there is an edge in $\mathcal{F}$ from a vertex in $\Psi(v)$ to a vertex in $\Psi(w)$). $\square$

**Lemma 34.** *Given some $v, w \in \mathcal{H} \setminus (\top \cup \{s\})$ with $\Psi'(v) \neq \bot'$ then the existence of an edge in $\mathbb{G}$ from $\Psi(v)$ to $\Psi(w)$ implies the existence of an edge in $\mathbb{G}'$ from $\Psi'(v)$ to $\Psi'(w)$.*

*Proof.* By Lemma 32 we have that $\Psi'(v) = \Psi(v)$. Since there is an edge in $\mathbb{G}$ from $\Psi(v)$ to $\Psi(w)$ there exist vertices $x \in \Psi(v)$ and $y \in \Psi(w)$ such that $(x \to y) \in \mathcal{F}$. Since $\Psi(v) = \Psi'(v)$ and $s \notin \Psi'(v)$ we have $x \neq s$. Since there is no edge in $\mathcal{F}$ that goes into $s$ we must have $y \neq s$. If $x$ was in $\top$ then we would have $\Psi(v) = \Psi(x) = \top$ which contradicts the fact that $v \notin \top$. Similarly $y \notin \top$. We hence have that $x, y \neq z$. Hence, $(x \to y) \in J$ and $x, y \notin \{s, z\}$ so by Lemma 27 Item 2 we either have that $(x \to y) \in J'$ or that $x, y \in \bot'$. But if $x \in \bot'$, then since $x \in \Psi'(v)$ (since, by the above, $\Psi'(v) = \Psi(v)$) we have that $\Psi'(v) = \bot'$ which is a contradiction. So $(x \to y) \in J'$ and hence $(x \to y) \in \mathcal{F}'$. Since there is an edge in $\mathbb{G}$ that goes into $\Psi(w)$ we have that $\Psi(w) \neq \bot$ so by Lemma 31 we have that $\Psi(w) \subseteq \Psi'(w)$ so $y \in \Psi'(w)$.
Suppose, for contradiction, that $\Psi'(v) = \Psi'(w)$. We know that $\Psi'(v) \neq \bot'$ and hence that $\Psi'(w) \neq \bot'$. By Lemma 32 we hence have that $\Psi(w) = \Psi'(w) = \Psi'(v) = \Psi(v)$ which is a contradiction. We hence have that $\Psi'(v) \neq \Psi'(w)$.
We hence have (since $x \in \Psi(v) = \Psi'(v)$ and $y \in \Psi'(w)$ and $(x \to y) \in \mathcal{F}'$) that there is an edge in $\mathbb{G}'$ from $\Psi'(v)$ to $\Psi'(w)$. $\square$

We have now proved Proposition 19: Item (a) is Lemma 29, Item (b) is Lemma 32 and Item (c) comes directly from lemmas 33 and 34. $\blacksquare$

# F  Proof of Optimality for Trees

In this section we prove that `0-Ising` and `longest-path` are optimal graph label prediction algorithms on trees in the sense of [1, Theorem 1]. We note that the `0-Ising` strategy when restricted to a tree was already proved optimal in [1] where it was called "`Halving`." Our proof

of optimality of `longest-path` uses much of "proof technology" from [1] so for the convenience of the reader in the next subsection we recall their notation and definitions.

## F.1 Ingredients from [1, Section 2]

Given a set $L$ of edge-disjoint paths contained in a tree $T$, we say that $l \in L$ is a grafted path if one of the two terminal vertices of $l$ is also an internal vertex of another path $l' \in L$. This shared vertex is called the graft vertex of $l$. We say that $L$ is a connected blanket if:

1. The union of all paths in $L$ forms a (connected) tree.

2. Every vertex in this (connected) tree can be an internal vertex of at most one such path.

3. Every grafted path in $L$ shares with the remaining paths in $L$ no vertices but the graft.

Finally, $L$ is a blanket if it is either a connected blanket or it has been obtained by a connected blanket after removing one or more of its paths. The size of a blanket $L$ is the number of its paths $|L|$. Note that a blanket need not include all edges of the original tree $T$. Also, observe that for any size $K < n$, a size-$K$ blanket over a tree $T$ always exists: take $L$ to be any set of $K$ distinct edges in $T$; then no paths of $L$ have internal vertices and the blanket property trivially holds. On the other hand, a given tree $T$ clearly admits many size-$K$ blankets. Let $\mathcal{L}(T, K)$ be the set of all size-$K$ blankets over $T$, and define the function LB (lower bound) as follows:

$$\text{LB}(T, K) := \max_{L \in \mathcal{L}(T,K)} \sum_{l \in L} \lfloor \log_2(|l|) \rfloor \tag{5}$$

where $|l|$ is the number of vertices in $l$. We state the lower bound for any graph label prediction algorithm proved in [1].

**Theorem 35.** *[1, Theorem 1], Given a tree $T$ and a number $K \in \mathbb{N}$, then for any online prediction algorithm $\mathcal{A}$ there exists a $\{0, 1\}$ labelling, $\nu$, of $T$ with cutsize at most $K$, on which algorithm $\mathcal{A}$ makes at least $\text{LB}(T, K)$ mistakes.*

## F.2 Proof of Optimality

Let $X$ be the set of cut edges of $T$ and let $K := |X|$. A subtree $Q$ is called a 2-tree (resp. 1-tree) if it is a subtree of $T$ with an inner boundary of two vertices (resp. one vertex) and all vertices in its inner boundary are leaves of it. Given a 1-tree or 2-tree $Q$ we let $Q^\circ$ be equal to $Q$ minus its inner boundary. We have the following lemma.

**Lemma 36.** *We can find a set $\mathbb{S}$, of 1-trees, and a set $\mathbb{T}$, of 1-trees and 2-trees, which satisfy the following.*

1. *Any 1-tree in $\mathbb{T}$ has a single edge and that edge is cut.*

2. *The trees in $\mathbb{S} \cup \mathbb{T}$ are edge-disjoint.*

3. *The union of the edges of the trees in $\mathbb{S} \cup \mathbb{T}$ is equal to the edge set of $T$.*

4. *$|\mathbb{T}| \leq 3K$*

5. *For every edge $(v, w)$ in $X$ we have a tree in $\mathbb{T}$ which has $(v, w)$ as a single edge. Note that this implies that $|\mathbb{T}| \geq K$*

6. *Given a tree $Q \in \mathbb{S}$, where $\partial_0(Q) = \{v\}$ for some $v \in T$, then we have $v \in \partial_0(R)$ for some $R \in \mathbb{T}$.*

7. *For any tree $Q \in \mathbb{T} \cup \mathbb{S}$ we have that $Q^\circ$ is identically labelled (i.e. there is no edge of $Q^\circ$ that is in $X$.)*

8. *For any tree $Q \in \mathbb{S}$ we have that $Q$ is identically labelled (i.e. there is no edge of $Q$ that is in $X$.)*

9. *For any trees $R, Q \in \mathbb{S} \cup \mathbb{T}$ we have that no vertex in the inner boundary of $Q$ is in $R^\circ$.*

*Proof.* Note first that in the following proof we may create subtrees containing a single vertex - such trees can be discarded. We prove by induction on $K$. For the base case $K = 1$ let $\{(v, w)\} := X$. Let $Q$ be the tree containing the single edge $(v, w)$ (Note that $Q^\circ$ has at most one vertex and is hence identically labelled). Let $\mathbb{A}$ be equal to the set of 1-trees with inner boundary $\{v\}$ or $\{w\}$ such that the trees in $\mathbb{A} \cup \{Q\}$ are edge-disjoint and the union of edges of the trees in $\mathbb{A} \cup \{Q\}$ is equal to the edge set of $T$. We then have $\mathbb{S} := \mathbb{A}$ and $\mathbb{T} := \{Q\}$. It is easy to check that all the statements of the lemma hold in this case.

Suppose that the inductive hypothesis holds for $K = \kappa$. We now consider the case that $K = \kappa + 1$: In this case choose an edge $e \in X$ and define $X' = X \setminus \{e\}$. Since $|X'| = \kappa$ we can find, by the inductive hypothesis, sets $\mathbb{S}'$ and $\mathbb{T}'$ to be equal to $\mathbb{S}$ and $\mathbb{T}$ (respectively) in the lemma if the set of cut edges was equal to $X'$ (instead of $X$). Let $S$ be the (unique) tree in $\mathbb{S}' \cup \mathbb{T}'$ that contains the edge $e$. Note that if $S$ was a 1-tree in $\mathbb{T}'$ then by Lemma 36 Item 1 we would have that $S$ had a single edge and that edge would be in $X'$ and hence not equal to $e$ which would be a contradiction. Hence, if $S$ is in $\mathbb{T}'$ then it has two inner boundary vertices. We hence have three cases.

1. $S \in \mathbb{S}'$: In this case let $(v, w) := e$ where $v$ is closer than $w$ to the inner-boundary vertex of $S$. Let $S'$ be the maximal subtree of $S$ with leaf $v$ that does not contain $w$. Let $Q$ be the tree containing the single edge $(v, w)$ (Note that $Q^\circ$ has at most one vertex and is hence identically labelled). Let $\mathbb{A}$ be equal to the set of 1-trees with inner boundary $\{v\}$ or $\{w\}$ such that the trees in $\mathbb{A} \cup \{S', Q\}$ are edge-disjoint and the union of edges of the trees in $\mathbb{A} \cup \{S', Q\}$ is equal to the edge set of $S$. Let $\mathbb{T} := \mathbb{T}' \cup \{S', Q\}$ (noting that by the inductive hypothesis we have $|\mathbb{T}| = 2 + |\mathbb{T}'| \le 2 + 3\kappa < 3(\kappa + 1) = 3K)$ and $\mathbb{S} := (\mathbb{S}' \setminus \{S\}) \cup \mathbb{A}$. By the inductive hypothesis (i.e. the conditions on $\mathbb{S}'$ and $\mathbb{T}'$) it is easy to check that all the statements of the lemma hold in this case.

2. $S \in \mathbb{T}'$ and $e$ is on the path between the inner boundary vertices of $S$: In this case let $(v, w) := e$. Let $S'$ (resp. $S''$) be the maximal subtree of $S$ with leaf $v$ (resp. $w$) that does not contain $w$ (resp. $v$). Let $Q$ be the tree containing the single edge $(v, w)$ (Note that $Q^\circ$ has at most one vertex and is hence identically labelled). Let $\mathbb{A}$ be equal to the set of 1-trees with inner boundary $\{v\}$ or $\{w\}$ such that the trees in $\mathbb{A} \cup \{S', S'', Q\}$ are edge-disjoint and the union of edges of the trees in $\mathbb{A} \cup \{S', S'', Q\}$ is equal to the edge set of $S$. Let $\mathbb{T} := (\mathbb{T}' \setminus \{S\}) \cup \{S'', S', Q\}$ (noting that by the inductive hypothesis we have $|\mathbb{T}| := 2 + |\mathbb{T}'| \le 2 + 3\kappa < 3(\kappa + 1) = 3K)$ and $\mathbb{S} := \mathbb{S}' \cup \mathbb{A}$. By the inductive hypothesis (i.e. the conditions on $\mathbb{S}'$ and $\mathbb{T}'$) it is easy to check that all the statements of the lemma hold in this case.

3. $S \in \mathbb{T}'$ and $e$ is not on the path between the inner boundary vertices of $S$: In this case let $\{x, y\}$ be the inner boundary of $S$ and let $(v, w) := e$. Let $z$ be the vertex where the path from $v$ to $y$ first meets the path from $x$ to $y$. Without loss of generality let $v$ be closer to $z$ than $w$ is. Let $S'$ (resp. $S''$) be the maximal subtree of $S$ with $x$ and $z$ (resp. $y$ and $z$) as leaves. Let $R$ be the maximal subtree of $S$ that has leaves $z$ and $v$. Let $Q$ be the tree containing the single edge $(v, w)$ (Note that $Q^\circ$ has at most one vertex and is hence identically labelled). Let $\mathbb{A}$ be equal to the set of 1-trees with inner boundary $\{v\}$ or $\{w\}$ or $\{z\}$ such that the trees in $\mathbb{A} \cup \{S', S'', Q, R\}$ are edge-disjoint and the union of edges of the trees in $\mathbb{A} \cup \{S', S'', Q, R\}$ is equal to the edge set of $S$. Let $\mathbb{T} := (\mathbb{T}' \setminus \{S\}) \cup \{S'', S', Q, R\}$ (noting that by the inductive hypothesis we have $|\mathbb{T}| := 3 + |\mathbb{T}'| \le 3 + 3\kappa = 3(\kappa + 1) = 3K)$ and $\mathbb{S} := \mathbb{S}' \cup \mathbb{A}$. By the inductive hypothesis (i.e. the conditions on $\mathbb{S}'$ and $\mathbb{T}'$) it is easy to check that all the statements of the lemma hold in this case.

$\square$

Let $\mathbb{S}$ and $\mathbb{T}$ be as in the above lemma. Let $J := \bigcup\{\partial_0(S) : S \in \mathbb{S} \cup \mathbb{T}\}$. Given $v \in J$ let $\mathbb{S}(v)$ be the set of trees in $\mathbb{S}$ that have an inner boundary of $\{v\}$ and let $\mathbb{T}(v)$ be the set of trees in $\mathbb{T}$ that have $v$ in their inner boundary.

**Lemma 37.** *We have* $|J| \le 6K$

*Proof.* By Lemma 36 Item 6 we have that any vertex in $J$ is in the inner boundary of a tree in $\mathbb{T}$ and there are (as for all $Q \in \mathbb{T}$ we have that $Q$ has an inner boundary of cardinality of at most two) at most $2|\mathbb{T}|$ such vertices. The result then follows by Lemma 36 Item 4 ☐

**Lemma 38.** *Suppose we have some $v \in J$. Then after we have received at least one example in $R^\circ$ for $\alpha$ of the trees $R \in \mathbb{S}(v)$, where $\alpha := |\mathbb{T}(v)| + 1$, we will no longer make any mistakes on any of the trees in $\mathbb{S}(v)$.*

*Proof.* Let $y$ be the true labelling of $T$ and let $y(v)$ denote the label of vertex $v$. Without loss of generality assume that $y(v) = 1$. We first note that by Lemma 36 Item 8 we have, for all $R \in \mathbb{S}(v)$, that every vertex $a$ in $R$ satisfies $y(a) = 1$ (since $v \in R$).

Suppose we have received at least one label in $R^\circ$ for $\alpha$ of the trees $R \in \mathbb{S}(v)$, where $\alpha := |\mathbb{T}(v)|+1$. Then let $\mathbf{u}$ be a consistent (with the observed examples) labelling of $T$ that minimises the cut. Given some $Q \in \mathbb{T}(v)$, define $z_Q$ to be the vertex in $Q$ that is adjacent to $v$. Let $C := \{(v, z_Q) : Q \in \mathbb{T}(v)\} \cup \bigcup\{E(R) : R \in \mathbb{S}(v)\}$. Then the restriction of $\mathbf{u}$ to the vertices in the edges in $C$ minimises the cutsize in $C$ given the observed examples and the labels $\{u_{z_Q} : Q \in \mathbb{T}(v)\}$.

By labelling all the vertices in $\bigcup\{V(R) : R \in \mathbb{S}(v)\}$ 1 we get a cut in $C$ of size no greater than $|\mathbb{T}(v)|$. But if $u_v = 0$ we get a cut in each of the trees in $\mathbb{S}(v)$ for which we have observed a label in, giving us a cut in $C$ of size at least $|\mathbb{T}(v)| + 1$. To minimise the cutsize in $C$ we must hence have that $u_v = 1$.

Hence, given a tree $R \in \mathbb{S}(v)$, the restriction of $\mathbf{u}$ to $R$ minimises the cutsize in $R$ given the observed labels and conditioned on $u_v = 1$. This cutsize is 0 if and only if all vertices in $R$ are labelled 1. Hence, $u_w = 1$ for all vertices $w \in V(R)$, so no mistake will be made in $R$. ☐

Let $\mathbb{U}$ be the set of trees $R \in \mathbb{S}$ in which a mistake is made in $R^\circ$.

**Lemma 39.** *We have $|\mathbb{U}| \leq 12K$*

*Proof.* By Lemma 38 we have that:

$$\mathbb{U} \leq \sum_{v \in J}(|\mathbb{T}(v)| + 1) = |J| + \sum_{v \in J} |\mathbb{T}(v)| = |J| + 2|\mathbb{T}| \tag{6}$$

where the last equality comes from the fact that for each tree $S$ in $\mathbb{T}$ we have that $S$ appears in at most two of the sets in $\{\mathbb{T}(v) : v \in J\}$ as it has at most two vertices in its inner boundary. By Lemma 37 and Lemma 36 Item 4 the result follows. ☐

**Lemma 40.** *Given a tree $S \in \mathbb{T} \cup \mathbb{U}$, the number of mistakes made in $S^\circ$ by the* `longest-path` *and* `0-Ising` *strategies is bounded above by $3\log_2(D(S) + 1)) + 5$.*

*Proof.* Let $y$ be the true labelling of $T$. Define a labelling, $\hat{y}$, of $S$ to be as follows. For $v \in \partial_0(S)$ we have $\hat{y}(v) := 0$. For $v \in S \setminus \partial_0(S)$ we have $\hat{y}(v) := 1$. Define the *full-algorithm* to be the algorithm run on $T$ with labelling $y$. Define the *sub-algorithm* to be the algorithm run on $S$ with labelling $\hat{y}$.

Since, by Lemma 36 Item 7, $S^\circ$ is identically labelled, by Theorem 4 the number of mistakes made by the full-algorithm in $S^\circ$ is no greater than the maximum number of mistakes, $M$, that the sub-algorithm makes in $S \setminus \partial_0(S)$ after it has received the labels on $\partial_0(S)$. We hence consider the sub-algorithm.

Since the cutsize of $\hat{y}$ is no greater than 2, $M \leq M_1 + M_2 + 3$ where $M_i$ is the number of mistakes made by the sub-algorithm in the PQ game of cutsize $i$. Let $\mathbb{G}_1$ (resp. $\mathbb{G}_2$) be the PQ graph at the start of the PQ game at cutsize 1 (resp. cutsize 2 (in the case that $S$ is a 2-tree)). By Theorem 8 there exists 1 (resp. 2) edge disjoint paths $p$ (resp. $p$, $q$) in $\mathbb{G}_1$ (resp. $\mathbb{G}_2$) such that $M_1 \leq 1 + \log_2(|p|)$ (resp. $M_2 \leq 1 + \log_2(|p|) + \log_2(|q|)$). But $|p| \leq D(S) + 1$ (resp. $|p|, |q| \leq D(S) + 1$). We hence have that $M_1 \leq 1 + \log_2(D(S) + 1)$ (resp. $M_2 \leq 1 + 2\log_2(D(S) + 1)$)

We hence have that $M \leq 3\log_2(D(S)) + 1) + 5$ which, by the above, is an upper bound on the number of mistakes made (by the full algorithm) in $S^\circ$. ☐

**Lemma 41.** *The number of mistakes, $\mathcal{M}$, made by the* `longest-path` *and* `0-Ising` *strategies are bounded above by:*

$$\mathcal{M} \leq \sum_{S \in \mathbb{T} \cup \mathbb{U}} 14 \log_2(D(S) + 1)) \tag{7}$$

*Proof.* Given a tree $S \in \mathbb{S} \cup \mathbb{T}$ let $M(S)$ be the number of mistakes made in $S^\circ$. By Lemma 36 Item 3 and the definition of $J$ every vertex in $T$ is either in $J$ or in $S^\circ$ for some $S \in \mathbb{S} \cup \mathbb{T}$ and hence the number of mistakes made in $T$ is upper bounded by:

$$\mathcal{M} \leq |J| + \sum_{S \in \mathbb{T} \cup \mathbb{S}} M(S) \tag{8}$$

$$= |J| + \sum_{S \in \mathbb{T} \cup \mathbb{U}} M(S) \tag{9}$$

$$\leq |J| + \sum_{S \in \mathbb{T} \cup \mathbb{U}} (3 \log_2(D(S) + 1)) + 5 \tag{10}$$

$$\leq 6K + \sum_{S \in \mathbb{T} \cup \mathbb{U}} (3 \log_2(D(S) + 1)) + 5 \tag{11}$$

$$\leq \sum_{S \in \mathbb{T} \cup \mathbb{U}} (3 \log_2(D(S) + 1)) + 11 \tag{12}$$

$$\leq \sum_{S \in \mathbb{T} \cup \mathbb{U}} 14 \log_2(D(S) + 1) \tag{13}$$

where Equation 9 comes from the definition of $\mathbb{U}$ (i.e. for all trees $S \in \mathbb{S} \setminus \mathbb{U}$ we have $M(S) = 0$), Equation 10 comes from Lemma 40, Equation 11 comes from Lemma 37 and Equation 12 comes from Lemma 36 Item 5. $\qquad\square$

We now define the following paths.

**Definition 42.** *For any 1-tree $S \in \mathbb{U} \cup \mathbb{T}$ define $\rho(S)$ to be a path in $S$ containing its inner boundary vertex that has maximum length. For any 2-tree $S \in \mathbb{T}$ let $\lambda(S)$ be the path between the inner boundary vertices of $S$. Let $\lambda'(S)$ be a path in $S$ that has a leaf in $\lambda(S)$ and is edge disjoint from $\lambda(S)$ that has maximum length. Let $\rho(S) = \mathrm{argmax}_{p \in \{\lambda(S), \lambda'(S)\}} |p|$.*

For the following we define the constant $\alpha := \log_2(4/3) / \log_2(2)$.

**Lemma 43.** *For any tree $S \in \mathbb{U} \cup \mathbb{T}$ we have $\log_2(|\rho(S)|) \geq \log_2(\frac{1}{3} D(S) + 1)$ which is bounded below by $\alpha \log_2(D(S) + 1)$.*

*Proof.* Direct from the definition of $\rho(S)$. $\qquad\square$

**Lemma 44.** $\{\rho(S) : S \in \mathbb{U} \cup \mathbb{T}\}$ *is a blanket of cardinality at most* $15K$.

*Proof.* We have, by Lemma 36 items 2, 3, 6 and 9, that $\{\rho(S) : S \in \mathbb{U}\} \cup \{\lambda(S) : S \in \mathbb{T}\} \cup \{\lambda'(S) : S \in \mathbb{T}\}$ is a connected blanket. So since $\{\rho(S) : S \in \mathbb{U} \cup \mathbb{T}\}$ is a subset of $\{\rho(S) : S \in \mathbb{U}\} \cup \{\lambda(S) : S \in \mathbb{T}\} \cup \{\lambda'(S) : S \in \mathbb{T}\}$ it is a blanket. The cardinality of $\{\rho(S) : S \in \mathbb{U} \cup \mathbb{T}\}$ follows from Lemma 36 Item 4 and Lemma 39. $\qquad\square$

We now define the following blanket.

**Definition 45.** *For $i \in \mathbb{N}_K$ inductively define:*

$$S_i = \mathrm{argmax}_{S \in (\mathbb{U} \cup \mathbb{T}) \setminus \{S_j : j < i\}} (|\rho(S)|) \tag{14}$$

*and define $\mathcal{B} := \{\rho(S_i) : i \in \mathbb{N}_K\}$.*

**Lemma 46.** $\mathcal{B}$ *is a blanket of size $K$ which satisfies:*

$$\sum_{p \in \mathcal{B}} \log_2(|p|) \geq \frac{\alpha}{15} \sum_{S \in \mathbb{U} \cup \mathbb{T}} \log_2(D(S) + 1) . \tag{15}$$

*Proof.* By Lemma 44 $\mathcal{B}$ is a subset of a blanket and is hence a blanket. Since $\mathcal{B}$ has $K$ elements it has, by Lemma 44 at least $\frac{1}{15}$th of the elements of $\{\rho(S) : S \in \mathbb{U} \cup \mathbb{T}\}$. So, since in forming $\mathcal{B}$ we picked the paths of greatest cardinality, we must have that $\sum_{p \in \mathcal{B}} \log_2(|p|) \geq \frac{1}{15} \sum_{p \in \{\rho(S): S \in \mathbb{U} \cup \mathbb{T}\}} \log_2(|p|) = \frac{1}{15} \sum_{S \in \mathbb{U} \cup \mathbb{T}} |\rho(S)|$ which, by Lemma 43, is at least $\frac{\alpha}{15} \sum_{S \in \mathbb{U} \cup \mathbb{T}} \log_2(D(S) + 1)$. $\square$

**Theorem 47.** *The number of mistakes $\mathcal{M}$ incurred by the* `longest-path` *and* `0-Ising` *strategies on a tree are bounded above by:*

$$\frac{210}{\alpha} \sum_{p \in \mathcal{B}} \log_2(|p|) \tag{16}$$

*where $\alpha := \log_2(4/3)/\log_2(2)$. So since $\mathcal{B}$ is a blanket of size $K$ the algorithm is, up to a constant factor, optimal (by Theorem 35).*

*Proof.* Direct from Lemmas 41 and 46. $\square$

# G Computing the predictions of the `0-Ising` strategy is NP-hard

**Theorem 48.** *Computing the predictions of the* `0-Ising` *strategy (see equation* (1)*) is NP-hard.*

## G.1 Proof of Theorem 48

NB: Whenever we mention "graph" in this section we mean a graph with source (label "0") and target (label "1") vertices.

Given a graph $\mathcal{J}$, let $\bot(\mathcal{J})$ and $\top(\mathcal{J})$ be the source and target vertices of $\mathcal{J}$ respectively. We define a *label consistent labeling* of $\mathcal{J}$ to be a labeling $\boldsymbol{u}$ of $\mathcal{J}$ such that $u_{\bot(\mathcal{J})} = 0$ and $u_{\top(\mathcal{J})} = 1$. We define the *cutsize* of $\mathcal{J}$ to be the minimum cutsize of a label consistent labelling of $\mathcal{J}$. We define $\mathcal{Z}(\mathcal{J})$ to be the number of label consistent labellings of $\mathcal{J}$ that have cutsize equal to the cutsize of $\mathcal{J}$.

In this proof we assume that we have an oracle (i.e. a black-box that takes constant time) $\mathbf{test}(\cdot, \cdot)$ that, given an input graph $\mathcal{J}$, of cutsize $K$, and an input vertex $z \in V(\mathcal{J})$, outputs "0" if there are fewer label consistent labelings of cutsize $K$ that label $z$ as "0" than those that label $z$ as "1" and outputs "1" otherwise. From $\mathbf{test}(\cdot, \cdot)$ we will construct a polynomial time algorithm for counting the number, $\mathcal{Z}(\mathcal{G})$, of label-consistent minimum cuts in a graph $\mathcal{G}$. Since the task of counting the number of label-consistent minimum cuts is #P-hard [10], we hence have that $\mathbf{test}(\cdot, \cdot)$ (i.e. computing equation (1)) is NP-hard. Let $n$ be the number of vertices in $\mathcal{G}$.

**Definition 49.** *Given a graph $\mathcal{C}$ with cutsize $L \leq n$ we define the graph $\mathcal{C}^*$ as follows:*

1. *There exists a set $X \subseteq V(\mathcal{C}^*)$ of $n + 1 - L$ vertices such that $X \cap V(\mathcal{C}) = \emptyset$ and $V(\mathcal{C}^*) = X \cup V(\mathcal{C})$*

2. *$\bot(\mathcal{C}^*) = \bot(\mathcal{C})$ and $\top(\mathcal{C}^*) = \top(\mathcal{C})$*

3. *$E(\mathcal{C}^*) = E(\mathcal{C}) \cup \{(\bot(\mathcal{C}), x) : x \in X\} \cup \{(x, y) : x, y \in X, x \neq y\} \cup \{(x, \top(\mathcal{C})) : x \in X\}$*

**Lemma 50.** *Given a graph $\mathcal{C}$ with cutsize $L \leq n$ the graph $\mathcal{C}^*$ has cutsize $n+1$ and $\mathcal{Z}(\mathcal{C}^*) = 2\mathcal{Z}(\mathcal{C})$.*

*Proof.* Since none of the vertices on the edges of $\{(\bot(\mathcal{C}), x) : x \in X\} \cup \{(x, y) : x, y \in X, x \neq y\} \cup \{(x, \top(\mathcal{C})) : x \in X\}$ are in $V(\mathcal{C}) \setminus \{\bot(\mathcal{C}), \top(\mathcal{C})\}$ we have that for any min-cut (and label consistent) labelling of $\mathcal{C}^*$, the restriction of that labelling onto $\mathcal{C}$ has cutsize $L$. So suppose we have a (label consistent) labelling $\boldsymbol{u}$ of $\mathcal{C}$ with cutsize $L$. We now extent to a labelling $\boldsymbol{u}'$ of $\mathcal{C}^*$.

If $u'_x := 1$ for every $x \in X$ we clearly have a cutsize of $n + 1$ ($L$ cuts in $E(\mathcal{C})$, $n + 1 - L$ cuts in $\{(\bot(\mathcal{C}), x) : x \in X\}$ and no cuts in $\{(x, y) : x, y \in X, x \neq y\} \cup \{(x, \top(\mathcal{C})) : x \in X\}$). If $u'_x := 0$ for every $x \in X$ we also have a cutsize of $n + 1$ ($L$ cuts in $E(\mathcal{C})$, $n + 1 - L$ cuts in $\{(x, \top(\mathcal{C})) : x \in X\}$ and no cuts in $\{(\bot(\mathcal{C}), x) : x \in X\} \cup \{(x, y) : x, y \in X, x \neq y\}$).

Now suppose that the above two conditions don't hold: i.e. that we have vertices $x, y \in X$ with $u_x := 0$ and $u_y := 1$. Then we have at least one cut in $\{(x, y) : x, y \in X, x \neq y\}$. Let $X_1 := \{x \in X : u_x = 1\}$ and let $X_0 := \{x \in X : u_x = 0\}$. Then we have $|X_1|$ cuts in $\{(\bot(\mathcal{C}), x) : x \in X\}$ and $|X_0|$ cuts in $\{(x, \top(\mathcal{C})) : x \in X\}$ giving a total of at least $n + 1 - L$ cuts in the union of these two sets. Adding the $L$ cuts in $V(\mathcal{C})$ gives us a total of over $n + 1$ cuts.

So the cutsize of $\mathcal{C}^*$ is $n + 1$ and moreover every (label consistent) labelling $\boldsymbol{u}$ of $\mathcal{C}$ of cutsize $L$ extends to exactly two labellings of $\mathcal{C}^*$ of minimum cutsize. Hence we have that $\mathcal{Z}(\mathcal{C}^*) = 2\mathcal{Z}(\mathcal{C})$. $\qquad \square$

**Definition 51.** *Given two graphs $\mathcal{C}$ and $\mathcal{D}$, define the* merger graph, $[\mathcal{C}, \mathcal{D}]$, *as follows:*

1. *The structure of $[\mathcal{C}, \mathcal{D}]$ is the graphs $\mathcal{C}$ and $\mathcal{D}$ with $\top(\mathcal{C})$ and $\bot(\mathcal{D})$ merged into a single vertex. i.e. we have a new vertex $z$ such that $V([\mathcal{C}, \mathcal{D}]) := (V(\mathcal{C}) \setminus \{\top(\mathcal{C})\}) \cup (V(\mathcal{D}) \setminus \{\bot(\mathcal{D})\}) \cup \{z\}$ and $E([\mathcal{C}, \mathcal{D}]) = (E(\mathcal{C}) \setminus \{(v, \top(\mathcal{C})) : v \in V(\mathcal{C})\}) \cup \{(v, z) : (v, \top(\mathcal{C})) \in E(\mathcal{C})\} \cup (E(\mathcal{D}) \setminus \{(\bot(\mathcal{D}), v) : v \in V(\mathcal{D})\}) \cup \{(z, v) : (\bot(\mathcal{D}), v) \in E(\mathcal{D})\}$.*

2. $\bot([\mathcal{C}, \mathcal{D}]) := \bot(\mathcal{C})$

3. $\top([\mathcal{C}, \mathcal{D}]) := \top(\mathcal{D})$

**Lemma 52.** *Given graphs $\mathcal{C}$ and $\mathcal{D}$ of cutsize $n + 1$ we have the following:*

1. $[\mathcal{C}, \mathcal{D}]$ *has cutsize $n + 1$*

2. $\mathcal{Z}([\mathcal{C}, \mathcal{D}]) = \mathcal{Z}(\mathcal{C}) + \mathcal{Z}(\mathcal{D})$.

3. *Given that $z$ is the vertex formed from the merger of $\top(\mathcal{C})$ and $\bot(\mathcal{D})$ then the output of $\textbf{test}(z, [\mathcal{C}, \mathcal{D}])$ is equal to $0$ if $\mathcal{Z}(\mathcal{C}) < \mathcal{Z}(\mathcal{D})$ and equal to $1$ otherwise.*

*Proof.* Let $z$ be the vertex formed from the merger of $\top(\mathcal{C})$ and $\bot(\mathcal{D})$. Suppose we have a (label consistent) labelling, $\boldsymbol{u}$, of $[\mathcal{C}, \mathcal{D}]$. If $u_z = 0$ (resp. $u_z = 1$) then there are at least $n + 1$ cuts in the edges $(E(\mathcal{D}) \setminus \{(\bot(\mathcal{D}), v) : v \in V(\mathcal{D})\}) \cup \{(z, v) : (\bot(\mathcal{D}), v) \in E(\mathcal{D})\}$ (resp. $(E(\mathcal{C}) \setminus \{(v, \top(\mathcal{C})) : v \in V(\mathcal{C})\}) \cup \{(v, z) : (v, \top(\mathcal{C})) \in E(\mathcal{C})\}$). Hence, we must have that $[\mathcal{C}, \mathcal{D}]$ has a cutsize of at least $n + 1$ and furthermore that $\boldsymbol{u}$ has cutsize $n + 1$ if and only if $u_x = 0$ for every $x \in V(\mathcal{C}) \setminus \{\top(\mathcal{C})\}$ (resp. $u_x = 1$ for every $x \in V(\mathcal{D}) \setminus \{\bot(\mathcal{D})\}$). So since there are $\mathcal{Z}(\mathcal{D})$ (resp. $\mathcal{Z}(\mathcal{C})$) labellings of $\{z\} \cup V(\mathcal{D}) \setminus \{\bot(\mathcal{D})\}$ (resp. $\{z\} \cup V(\mathcal{C}) \setminus \{\top(\mathcal{C})\}$) that label $z$ as $0$ and $\top(\mathcal{D})$ as $1$ (resp. label $z$ as $1$ and $\bot(\mathcal{C})$ as $0$) and have $n + 1$ cuts in the edges $(E(\mathcal{D}) \setminus \{(\bot(\mathcal{D}), v) : v \in V(\mathcal{D})\}) \cup \{(z, v) : (\bot(\mathcal{D}), v) \in E(\mathcal{D})\}$ (resp. $(E(\mathcal{C}) \setminus \{(v, \top(\mathcal{C})) : v \in V(\mathcal{C})\}) \cup \{(v, z) : (v, \top(\mathcal{C})) \in E(\mathcal{C})\}$) we have that there are exactly $\mathcal{Z}(\mathcal{D})$ (resp. $\mathcal{Z}(\mathcal{C})$) label consistent labellings of $[\mathcal{C}, \mathcal{D}]$ that label $z$ as $0$ (resp. $z$ as $1$) and have cutsize $n + 1$. All the items of the lemma follow. $\qquad \square$

**Definition 53.** *Given some $\alpha \in \mathbb{N}_n$ we define $\mathcal{Q}(\alpha)$ as follows:*

1. *There exists a set $A$ of $\alpha$ vertices such that $\bot(\mathcal{Q}(\alpha)), \top(\mathcal{Q}(\alpha)) \notin A$ and $V(\mathcal{Q}(\alpha)) = \{\bot(\mathcal{Q}(\alpha)), \top(\mathcal{Q}(\alpha))\} \cup A$.*

2. $E(\mathcal{Q}(\alpha)) = \{(\bot(\mathcal{Q}(\alpha)), v) : v \in A\} \cup \{(v, \top(\mathcal{Q}(\alpha))) : v \in A\}$

Given, for some $l$, the sequence $(\alpha_0, \alpha_1, ..., \alpha_l)$ with $\alpha_i < \alpha_{i+1} < n$ we now define a graph $\mathfrak{B}(\alpha_0, \alpha_1, ..., \alpha_l)$ that has a label consistent minimum cutsize of $n + 1$ and such that $\mathcal{Z}(\mathfrak{B}(\alpha_0, \alpha_1, ..., \alpha_l)) = 2 \sum_{i=0}^{l} 2^{\alpha_l}$

**Definition 54.** *Given, for some $l$, the sequence $(\alpha_0, \alpha_1, ..., \alpha_l)$ with $\alpha_i < \alpha_{i+1} < n$ we inductively define $\mathfrak{B}(\alpha_0, \alpha_1, ..., \alpha_l)$ as follows:*

1. $\mathfrak{B}(\alpha_0) = \mathcal{Q}(\alpha_0)^*$

2. $\mathfrak{B}(\alpha_0, \alpha_1, ..., \alpha_l) := [\mathcal{Q}(\alpha_0)^*, \mathfrak{B}(\alpha_1, \alpha_2..., \alpha_l)]$

**Lemma 55.** *Given, for some $l$, the sequence $(\alpha_0, \alpha_1, ..., \alpha_l)$ with $\alpha_i < \alpha_{i+1} \leq n$ we have that $\mathfrak{B}(\alpha_0, \alpha_1, ..., \alpha_l)$ has a cutsize of $n + 1$ and $\mathcal{Z}(\mathfrak{B}(\alpha_0, \alpha_1, ..., \alpha_l)) := 2 \sum_{i=0}^{l} 2^{\alpha_l}$.*

*Proof.* Noting that $\mathcal{Z}(\mathcal{Q}(\alpha_0)) = 2^{\alpha_0}$ and hence, by Lemma 50 $\mathcal{Z}(\mathcal{Q}(\alpha_0)^*) = 2 \cdot 2^{\alpha_0}$ and $\mathcal{Q}(\alpha_0)^*$ has cutsize $n + 1$, the proof is direct by induction on $l$ using items 1 and 2 of Lemma 52. $\qquad\square$

**Lemma 56.** $\mathcal{Z}(\mathcal{G}) \leq 2^n$

*Proof.* Note that there are $2^n$ labellings of $\mathcal{G}$ which implies the result. $\qquad\square$

The following algorithm calculates $\mathcal{Z}(\mathcal{G})$ (unless $\mathcal{Z}(\mathcal{G}) = 1$ in which case computing $\mathcal{Z}(\mathcal{G})$ is done by running $\mathbf{test}([\mathfrak{B}(0), \mathcal{G}^*], z)$ (where $z$ is the vertex formed from the merger of $\top(\mathfrak{B}(0))$ and $\perp(\mathcal{G}^*)$)).

**Algorithm 57.** *Throughout the algorithm we maintain a graph $\mathcal{J}$, which is equal to $\mathfrak{B}(\alpha_0, \alpha_1, ..., \alpha_l)$ for some $l$ and sequence $(\alpha_0, \alpha_1, ..., \alpha_l)$. $\mathcal{J}$ is initialised to be equal to $\mathfrak{B}(n)$. The algorithm loops over the following:*

1. *Let $\mathfrak{B}(\alpha_0, \alpha_1, ..., \alpha_l) := \mathcal{J}$. Construct the graph $[\mathcal{J}, \mathcal{G}^*]$. Let $z$ be the vertex in $[\mathcal{J}, \mathcal{G}^*]$ formed from the merger of $\top(\mathcal{J})$ and $\perp(\mathcal{G}^*)$.*

2. *Run $\mathbf{test}([\mathcal{J}, \mathcal{G}^*], z)$. If the output is 1 then set $\mathcal{J} \leftarrow \mathfrak{B}(\alpha_0 - 1, \alpha_1, \alpha_2, ..., \alpha_l)$. If the output is 0 then run the following:*

   (a) *Construct the graph $[[\mathfrak{B}(0), \mathcal{J}], \mathcal{G}^*]$. Let $z'$ be the vertex in $[[\mathfrak{B}(0), \mathcal{J}], \mathcal{G}^*]$ formed from the merger of $\top([\mathfrak{B}(0), \mathcal{J}])$ and $\perp(\mathcal{G}^*)$.*

   (b) *Run $\mathbf{test}([[\mathfrak{B}(0), \mathcal{J}], \mathcal{G}^*], z')$. If the output is 1 then the algorithm terminates outputting $\mathcal{Z}(\mathcal{G}) \leftarrow 1 + \sum_{i=0}^{l} 2^{\alpha_i}$. If, instead, the output is 0 we set $\mathcal{J} \leftarrow \mathfrak{B}(\alpha_0 - 1, \alpha_0, \alpha_1, ..., \alpha_l)$*

**Theorem 58.** *Given an oracle $\mathbf{test}(\cdot, \cdot)$, Algorithm 57 outputs $\mathcal{Z}(\mathcal{G})$ in polynomial time.*

*Proof.* Note first that by Lemma 50 and Lemma 52 Item 1 all graphs, $\mathcal{C}$, in the algorithm satisfy $\mathcal{Z}(\mathcal{C}) = n + 1$.

Since, by Lemma 56, we have that $\mathcal{Z}(\mathcal{G}) \leq 2^n$ we can find, for some $l, \beta_0, \beta_1, ..., \beta_l \in \mathbb{Z}$ such that $\beta_1 \geq 0$, $\beta_l < n$, for all $i$ we have $\beta_i < \beta_{i+1}$ and $\mathcal{Z}(\mathcal{G}) = 1 + \sum_{i=q}^{l} 2^{\beta_i}$. By Lemma 50 we have $\mathcal{Z}(\mathcal{G}^*) = 2\mathcal{Z}(\mathcal{G}) = 2\left(1 + \sum_{i=q}^{l} 2^{\beta_i}\right)$. Let $\mathcal{J}_t$ be the graph $\mathcal{J}$ at the start of the $t^{\text{th}}$ loop.

We prove, by reverse induction (i.e. from $l$ to 0) on $j \leq l$ that there exists a $t$ such that $\mathcal{J}_t = \mathfrak{B}(\beta_j, \beta_{j+1}, ..., \beta_l)$. We first show the base case: that there is some time $t$ such that $\mathcal{J}_t = \beta_l$. To see this suppose at some time $t'$ we have that $\mathcal{J}_{t'} = \mathfrak{B}(\beta')$ for some $\beta' > \beta_l$. We consider the $(t')^{\text{th}}$ loop. By Lemma 52 Item 3 and since $\mathcal{Z}(\mathcal{G}^*) \leq 2 \cdot 2 \cdot 2^{\beta_l} \leq 2 \cdot 2^{\beta'} = \mathcal{Z}(\mathcal{J}_{t'})$ (where the last equality comes from Lemma 55), the result of $\mathbf{test}([\mathcal{J}_{t'}, \mathcal{G}^*], z)$ is 1. We hence have that $\mathcal{J}_{t'+1} = \mathfrak{B}(\beta' - 1)$. Hence, since $\mathcal{J}_1 = \mathfrak{B}(n)$ and $n > \beta_l$ we have that $\mathcal{J}_{1+n-\beta_l} = \mathfrak{B}(\beta_l)$. We have hence proved that the inductive hypothesis holds for $j = l$ so now suppose it holds for some $0 < j \leq l$. We now show that it holds for $j - 1$. Since it holds for $j$ choose $t''$ such that $\mathcal{J}_{t''} = \mathfrak{B}(\beta_j, \beta_{j+1}, ..., \beta_l)$. We consider the $(t'')^{\text{th}}$ loop. By Lemma 52 Item 3 and since $\mathcal{Z}(\mathcal{G}^*) \geq 2\left(1 + \beta_0 + \sum_{i=j}^{l} 2^{\beta_i}\right) > 2\sum_{i=j}^{l} 2^{\beta_i} = \mathcal{Z}(\mathcal{J}_{t''})$ (where the last equality comes from Lemma 55), the result of $\mathbf{test}([\mathcal{J}_{t'}, \mathcal{G}^*], z)$ is 0. By Lemma 52 Item 3 and since $\mathcal{Z}(\mathcal{G}^*) \geq 2\left(1 + \beta_0 + \sum_{i=j}^{l} 2^{\beta_i}\right) > 2\left(1 + \sum_{i=j}^{l} 2^{\beta_i}\right) = \mathcal{Z}([\mathfrak{B}(0), \mathcal{J}_{t''}])$ (where the last equality comes from Lemma 55), the result of $\mathbf{test}([[\mathfrak{B}(0), \mathcal{J}_{t''}], \mathcal{G}^*], z')$ is 0. We hence have that $\mathcal{J}_{t''+1} = \mathfrak{B}(\beta_j - 1, \beta_j, \beta_{j+1}, ..., \beta_l)$. Note now that if $\beta_j - 1 = \beta_{j-1}$ we are done. Else we have (as $\beta_{j-1} < \beta_j$) that $\beta_j - 1 > \beta_{j-1}$. Now suppose we have some $t'$ with $\mathcal{J}_{t'} = \mathfrak{B}(\beta', \beta_j, \beta_{j+1}, ..., \beta_l)$ for some $\beta' > \beta_{j-1}$. We consider the $(t')^{\text{th}}$ loop. By Lemma 52 Item 3 and since $\mathcal{Z}(\mathcal{G}^*) \leq 2\left(2^{\beta_{j-1}} + \sum_{i=j-1}^{l} 2^{\beta_i}\right) = 2\left(2 \cdot 2^{\beta_{j-1}} + \sum_{i=j-1}^{l} 2^{\beta_i}\right) \leq 2\left(2^{\beta'} + \sum_{i=j-1}^{l} 2^{\beta_i}\right) = \mathcal{Z}(\mathcal{J}_{t'})$ (where the last equality comes from Lemma 55), the result of $\mathbf{test}([\mathcal{J}_{t'}, \mathcal{G}^*], z)$ is 1. We hence have that $\mathcal{J}_{t'+1} = \mathfrak{B}(\beta' - 1, \beta_j, \beta_{j+1}, ...\beta_l)$. Hence, since $\mathcal{J}_{t''+1} = \mathfrak{B}(\beta_j - 1, \beta_j, \beta_{j+1}, ..., \beta_l)$ and $\beta_j - 1 \geq \beta_{j-1}$ we have that $\mathcal{J}_{t''+\beta_j-\beta_{j-1}} = \mathfrak{B}(\beta_{j-1}, \beta_j, \beta_{j+1}, ..., \beta_l)$. This completes the proof of the inductive hypothesis.

By the above we have that there exists a time $t$ such that $\mathcal{J}_t = \mathfrak{B}(\beta_0, \beta_1, ..., \beta_l)$. We now show that the algorithm outputs at time $t$. By Lemma 52 Item 3 and since $\mathcal{Z}(\mathcal{G}^*) =$

$2\left(1+\sum_{i=0}^{l} 2^{\beta_i}\right) > 2\sum_{i=0}^{l} 2^{\beta_i} = \mathcal{Z}(\mathcal{J}_t)$ (where the last equality comes from Lemma 55), the result of $\mathbf{test}([\mathcal{J}_{t'}, \mathcal{G}^*], z)$ is 0. By Lemma 52 Item 3 and since $\mathcal{Z}(\mathcal{G}^*) = 2\left(1+\sum_{i=0}^{l} 2^{\beta_i}\right) = \mathcal{Z}([\mathfrak{B}(0), \mathcal{J}_{t''}])$ (where the last equality comes from Lemma 55), the result of $\mathbf{test}([[\mathfrak{B}(0), \mathcal{J}_{t''}], \mathcal{G}^*], z)$ is 1. The algorithm hence outputs at time $t$ with output $1 + \sum_{i=0}^{l} 2^{\beta_i}$ which is equal to $\mathcal{Z}(\mathcal{G})$. The algorithm hence outputs correctly.

We now show that given an oracle $\mathbf{test}(\cdot, \cdot)$, Algorithm 57 runs in in polynomial time. Note first that it is clear that each loop takes polynomial time. Hence, all that is required to show is that there is a polynomial number of loops. Let $\mathfrak{B}(\beta_0^t, \beta_1^t, ..., \beta_{l^t}) := \mathcal{J}_t$. It is clear that for all $t$ we have $\beta_0^{t+1} = \beta_0^t - 1$ and hence, since $\beta_0^1 = n$ we have at most $n+1$ loops. This completes the proof. $\square$

Since, with an oracle $\mathbf{test}(\cdot, \cdot)$, Algorithm 57 solves a #P-hard problem in polynomial time we must have that $\mathbf{test}(\cdot, \cdot)$ is NP-hard.