[Reviews · NeurIPS 2015]

Submitted by Assigned_Reviewer_1

Online binary vertex classification over a graph. In the problem, initially the learner is provided with a graph $G$ and on each round, possibly adversarially a node from this graph is picked and the learner provides a label in $\{0,1\}$ for this node. Nature then reveals the true label $y_t$ for this node. The prediction made by the learner is based on the ising model when the temperature parameter approaches 0. However this is a compitationally hard problem to solve and so the authors propose an approximate scheme based on constructing Picard-Queyranne graphs that preserve combinatorial structure of minimum cuts. Now the learner makes predictions by playing an online game on the PQ graph instead. The authors provide a mistake bound for the algorithm. In the case when the graph is a tree the mistake bounds are comparable to treeopt algortihm

I really liked the paper but it definitely was a hard read and the paper is not all that well written. Even the basic problem setup and description is staggered around in the first two sections. It would nice to give a bullet point type list to give a concise but complete description of the basic problem. The authors should also consider giving an informal proof outline and high level results in one of the first two sections. The 0-ising model is a very natural although intractable benchmark model to compare against. The comparison of the treeopt to the proposed algorithm is encouraging.The reduction to PG graphs and the prediciton PQ prediction game is interesting and novel

Overall I believe the work is defintiely worth publishing.
Summary: The problem considered is a very interesting one and has practical applications. The time complexity of the algorithm is order square of number of vertices which makes it suitable only for small to medium scale problems.

Submitted by Assigned_Reviewer_2

Summary: -------- The paper studies prediction of labels on a graph, where the nodes are data points, the edges indicate similarity in label, and getting different labels on the sides of an edge is penalized. The authors analyze the problem in an online setting, and provide a general mistake bound (Thm. 4) that holds for any algorithm that has three properties: monotonicity (observing a label can only result in more predictions of that label), permutation-invariance (order of observations does not matter), and a "Markov" property (predictions on a node are influenced only by its immediate connections). Next, the authors provide two prediction strategies that satisfy these properties (Thm. 9), one of which (0-Ising) is computationally inefficient while the other (longest-path) has polynomial running time in the size of the graph. Finally, the authors show a per-cluster mistake bound for these two strategies (Thm. 10) that, combined with the general mistake bound of Thm. 4, results in a concrete bound for these two algorithms. The new bound retrieves the previous optimal bound on trees, while it still gives meaningful (in cases improved) bounds when the graph is not a tree.

Quality and Clarity: -------------------- Generally, the paper is well-written: the setting, definitions, and logical steps are laid out and explained clearly, which makes the paper easy to read even for those who are not directly working in this area. There are some problems to fix, which I have mentioned at the end.

The theory seems to work, although I have not verified all the proofs in detail.

Originality and Significance: ----------------------------- The contribution that the paper lays out seems interesting and I think it will be useful for the community. However, I am not working in this area directly, so I leave a deeper comparison with the related work to other reviewers that are more familiar with the previous work in this particular area.

Other issues: -------------

- It is not specified what is the format of the max-flow returned in step 2 of Figure 1, which propagates to the definition of graph $$I$$ in step 3. In particular, in Figure 2(c), has 1->3->2 been in the max flow? If no, then why has 11->10->12 been there? If yes, then why are the edges 1-3 and 3-2 still in I? - On page 3, paragraph 2, line 7, please fix the definition of the set of edges of the quotient graph. - Perhaps the first sentence of Sec. 2 needs to be moved to the first page to become the first sentence of the last paragraph on that page. Then I think the phrase "this graph" on the third-to-last line on page 1 will have a close-by reference. - perhaps the references [3] and [4] on line 4 of the first paragraph are swapped.
Summary: This is a well-written paper. The theory seems to work and I think the contributions, as laid out in the paper, will be interesting to the community. However, I am not directly working in this specific area, so I leave it to the other reviewers to comment on the significance.

Submitted by Assigned_Reviewer_3

** Summary of paper

This paper presents a new online algorithm to predict label on a

graph. The graph labeling prediction problem is motivated by

semi-supervised learning where labeled and unlabeled are vertices

on a graph and edges represent closeness of these data. This work

uses an Ising model and seeks to optimize a bound on number of

mistake made by the learner online given a constraint on the

complexity of true labeling on the graph, such as the number of

edges connecting disagreeing vertex in the graph.

The paper exploits a transformation of the graph to Picard-Queyranne

graph and analyses the mistake bounds for two prediction strategies

via analysis of mistake bounds in PQ-games and per-cluster mistakes

bounds. The final mistake bounds are compared with mistake bounds

in existing literature.

** Quality

The final results reproduce the optimal result for online

labelling problem on trees. In comparison to mistake bounds in

existing literature, this paper's result is better when the graph

can be covered by label-consistent clusters of different

diameter. This is because the analysis is done per-cluster and this

should be a better bound in most natural cases.

** Clarity

This paper is well written. The background material required to

understand the PQ graphs is sufficiently covered.

** Originality

This paper makes novel use of PQ-graph and per-cluster analysis to

achieve the final mistake bounds.

** Significance

The algorithms proposed and theoretical results constitute significant

technical contributions.
Summary: This a high quality paper that makes novel use of a Picard-Queyranne graph to achieve new mistake bounds in the online graph label prediction problem. The new bounds are arguably better than existing ones for most natural graph labelling.

Submitted by Assigned_Reviewer_4

The paper discusses the problem of semi-supervised learning where data points are vertices on a graph and edges represent our belief that two vertices are likely to have the same label. The authors analyze a new method (longest-path) for predicting vertex labels incrementally (online) and characterize the number of mistakes made depending on the behaviour of the adversary (who chooses the next vertex and its label). The setup is assuming the labeling follows an underlying Ising distribution. The proposed method is analyzed with the help of a combinatorial structure (PQ graph) that collapses vertices of the same label. This allows for the development of a tractable algorithm (quadratic in the size of the graph) that avoids counting of the label cuts (sets of sedge connecting vertices with different labels).

Quality: The results and analysis appear to be correct. The proposed new algorithm has tractable runtime and improves the error bound in certain settings.

Clarity: I found the paper poorly structures and hard to follow. The introductory sections are well written and set the stage. However, the analysis does not follow smoothly. It is unclear where the paper is going and why the intermediate results on regular graphs are presented. Furthermore, the discussion section appears to give a lot of new information (e.g., summarizing the 4+1 proposed) methods for label prediction). The paper could benefit a lot by given all that information in the paper and not at the end to help evaluate the context and significance of this work.

Originality, Significance: The paper proposes a new algorithm that makes use of the PQ-graph to reduce complexity. The longest-path algorithm can achieve lower error bounds in certain cases, and maths the optimal bound on trees. Unfortunately, The paper is outside my area so my ability to judge its significance and originality is limited.

Summary: The paper discusses error bounds for a new algorithm for predicting incrementally the labels of a graph in a semi-supervised setting. I found the flow of the paper not very smooth and thus hard to follow and evaluate the significance of the results.

Author Feedback
Author rebuttal: Dear Reviewers, we would like to thank you for your comments. Below we respond to the specific comments of reviewers 1 and 3.

Reviewer 1:

- "It is not specified what is the format of the max-flow returned in step 2 of Figure 1, which propagates to the definition of graph $$I$$ in step 3. In particular, in Figure 2(c), has 1 - > 3 - > 2 been in the max flow? If no, then why has 11 - > 10 - > 12 been there? If yes, then why are the edges 1-3 and 3-2 still in I?"

* The max-flow F returned in step 2 is just the set of directed edges in a 0-1 max flow (Note generally there may be more than one such max flow). The logic of step 3 which creates graph I, is that all the directed edges of the max flow F are in I and every (undirected) edge in H is also in I if it does not have an orientation which is in F.

* Thus 1 - > 3 - > 2 is not in the max flow.

- ". . . then why has 11 - > 10 - > 12 been there?"

* Note that either "1 1- > 10 - > 12" or "11 - > 12" must be in the max flow but not both.

- ". . . then why are the edges 1-3 and 3-2 still in I?"

* By the definition of I since their orientations are not in the max flow they are included as undirected edges.

- On page 3, paragraph 2, line 7, please fix the definition of the set of edges of the quotient graph.

* The definition is now fixed.

- Perhaps the first sentence of Sec. 2 needs to be moved to the first page to become the first sentence of the last paragraph on that page. Then I think the phrase "this graph" on the third-to-last line on page 1 will have a close-by reference.

* Now fixed.

- perhaps the references [3] and [4] on line 4 of the first paragraph are swapped.

* Now swapped.

Reviewer 3:

- ". . . characterize the number of mistakes made depending on the behavior of the adversary (who chooses the next vertex and its label). The setup is assuming the labeling follows an underlying Ising distribution."

* The set-up is in fact completely adversarial, i.e., the labeling is chosen by the adversary.

- " It is unclear where the paper is going and why the intermediate results on regular graphs are presented."

* The structure of the mistake bound analysis is discussed in the beginning of Section 3 in lines 218-233.

* Note that there are no results presented on "regular graphs" however, an analysis is given of "regular graph label prediction algorithms" (permutation-invariant, label-monotone, Markov) is presented [Theorem 4]. The two algorithms considered in this paper longest-path and 0-Ising are both regular algorithms [Theorem 9]. [Theorem 10] in conjunction with [Theorem 4] completes the mistake bound analysis.